# Using Clause Predictions for Learning-Augmented Constraint Satisfaction

## Abstract

We continue a recent flourishing line of work on studying NP-hard problems with predictions and focus on fundamental constraint satisfaction problems such as Max-E3SAT and its weighted variant. Max-E3SAT is the natural 'maximizing' generalization of 3SAT, where we want to find an assignment to maximize the number of satisfied clauses. We introduce a clause prediction model, where each clause provides one noisy bit (accurate with probability $1/2 + \varepsilon$) of information for each variable participating in the clause, based on an optimal assignment. We design an algorithm with approximation factor of $7/8 + \Theta(\varepsilon^2/\log(1/\varepsilon))$. Our algorithm leverages the fact that in our model, high-occurrence variables tend to be highly predictable. By carefully incorporating a classic algorithm for Max-E3SAT with bounded-occurrence, we are able to bypass the worst-case lower bounds of $7/8$ without advice (assuming $P \neq NP$).

We also give hardness results of Max-E3SAT in other well studied prediction models such as the $\varepsilon$-label and subset prediction models of Cohen-Addad et al. (NeurIPS 2024) and Ghoshal et al. (SODA 2025). In particular, under standard complexity assumptions, in these prediction models, we show Max-E3SAT is hard to approximate to within a factor of $7/8 + \delta$ and Max-E3SAT with bounded-occurrence $B$ (every variable appears in at most $B$ clauses) is hard to approximate to within a factor of $7/8 + O(1/\sqrt{B}) + \delta$ for $\delta$ a specific function of $\varepsilon$. Our first lower bound result is based on the framework proposed by Ghoshal et al. (SODA 2025), and the second uses a randomized reduction from general instances of Max-E3SAT to bounded-occurrences instances proposed by Trevisan (STOC 2001).

## 1 Introduction

Learning-augmented algorithms are a popular recent paradigm for proving beyond worst case algorithmic results. This recent subfield is at the crossroads of algorithm design and machine learning, and is motivated by practical scenarios where it is possible to learn unknown information about the input at hand using ML tools, e.g. predictors trained on prior data or similar instances. This has found success in many algorithmic problems where uncertainty about the input causes mis-performance, such as inputs arriving in streaming or online settings (Lykouris & Vassilvitskii, 2021; Mitzenmacher & Vassilvitskii, 2022; Hsu et al., 2019; Jiang et al., 2020; Chen et al., 2022b), behavior of queries for datastructures (Mitzenmacher & Vassilvitskii, 2022; Kraska et al., 2018), or uncertainty of variables in optimization problems (Dinitz et al., 2021; Chen et al., 2022a; Cohen-Addad et al., 2024).

Beyond practical motivations, the viewpoint of learning augmented algorithms offers an alternative perspective in understanding 'what makes an algorithmic problem difficult?' Concretely, designing algorithms in the learning-augmented model requires defining a notion of 'natural predictions', which give noisy advice about the input, and understanding how to effectively use such advice in the algorithm design process to 'nudge' hard inputs to potentially more feasible instances.

We focus on a recent line of work on constraint satisfaction problems (CSPs) with advice and more broadly NP-Hard problems with advice (Ergun et al.; Gamlath et al., 2022; Antoniadis et al., 2024; Braverman et al., 2024; Cohen-Addad et al., 2024; Bampis et al., 2024; Ghoshal et al., 2025). A CSP is defined by a set of variables taking values from some domain, and a set of clauses (or constraints), where each clause is given as a predicate that involves a subset of the variables. We are primarily concerned with its optimization version: finding an assignment to the variables that maximizes the number of satisfied clauses. CSPs are fundamental because of their generality: they naturally model many other NP-hard optimization problems, and constitute an active line of research in both algorithms and complexity. For example, an examination scheduling problem can be formulated as a CSP by modeling each examination as a variable, its available time slots as the domain, and the conflicts between examinations as constraints.

Our paper focuses on arguably the simplest CSP which already faces a strong hardness barrier: Max-E3SAT. Specifically, we are given $n$ boolean variables and $m$ clauses, each clause is an OR of 3 boolean variables such as $(x_1 \lor x_2 \lor x_3)$. Our goal is to find an assignment of the boolean variables maximizing the number of satisfied clauses (see Definition 1.8). For this problem, it is known that the following very simple undergraduate level algorithm is already optimal (assuming $P \neq NP$ (Håstad, 2001)): just pick a uniformly random assignment without looking at any of the constraints! Because every clause has 3 variables, the probability that any individual clause is satisfied is $7/8$. Thus linearity of expectation implies we can always satisfy $7m/8$ clauses in expectation, obtaining an approximation factor of $7/8$. Again, to emphasize, this simple algorithm which does not look at the structure of the instance at all, is provably the best one can hope for in polynomial time in the worst-case (assuming $P \neq NP$).

Thus, due to the fundamental nature of the problem and the simplicity of the classical optimal solution, Max-E3SAT presents an intriguing challenge for the field learning augmented algorithms. The discussion motivates asking following natural questions:

*Question 1: What is a natural model of predictions for the Max-E3SAT problem?*

We note that usually in learning-augmented algorithm design, existing worst-case algorithms heavily inspire the augmented algorithm design process. This is because classical solutions point to an algorithmic structure that the algorithm designer can use, and (as an oversimplification), an augmented algorithm simply 'exploits' this structure better using predictions. For example, many augmented algorithm simply pick better parameters of standard algorithm (such as the starting spot in binary search (Lin et al., 2022; Fu et al., 2025), better distribution over actions in Ski-Rental (Purohit et al., 2018; Bamas et al., 2020), or a different sampling probability in streaming algorithms (Chen et al., 2022b)). However in the case of the fundamental Max-E3SAT problem, it seems difficult to use this meta approach since the known optimal algorithm described above uses absolutely no structure of the input instance! Thus we ask:

*Question 2: How can we use predictions to exploit the underlying structure of the Max-E3SAT input?*

Lastly, we state our main goal of obtaining better approximation results.

*Question 3: Can we obtain a better than a $7/8$ approximation (in polynomial time) under natural predictions? What are the fundamental limits of the approximation factor using predictions?*

In this paper we present progress towards all of the three fundamental questions.

## 1.1 OUR CONTRIBUTIONS AND DISCUSSION OF RESULTS

**Contribution Towards Question 1.** Towards the first question, there are three natural prediction models: Label Advice, Variable Subset Advice, and Clause Advice. First we state some notation before defining the advice models. We are given formula $\phi$ with clauses $C_1, C_2, \ldots, C_m$ and variables $x_1, x_2, \ldots, x_n$ (all

variables will be boolean, represented as $\pm 1$). Let clause $C_j$ consist of variables $x_{j_1}, \ldots, x_{j_s}$. Let $x^* = (x_1^*, \ldots, x_n^*)$ be a fixed optimal assignment. In the first two prediction models, we receive advice $\tilde{x} = (\tilde{x}_1, \ldots, \tilde{x}_n)$, where $\tilde{x}_i$ is a noisy prediction of $x_i$.

**Definition 1.1** (Label Advice). *In this model, $\tilde{x}_i = x_i^*$ with probability $(1 + \varepsilon)/2$ and $\tilde{x}_i = -x_i^*$ with probability $(1 - \varepsilon)/2$. Moreover, all $\tilde{x}_i$ are independent.*

**Definition 1.2** (Variable Subset Advice). *In this model, $\tilde{x}_i = x_i^*$ with probability $\varepsilon$ and $\tilde{x}_i = 0$ (null) with probability $1 - \varepsilon$. Moreover, all $\tilde{x}_i$ are independent.*

These two prediction models were defined in Cohen-Addad et al. (2024); Ghoshal et al. (2025) (in the context of other CSPs) and we introduce the third prediction model below in the context of CSPs. In the third prediction model, we receive advice $\tilde{C}_\phi = (\tilde{C}_1, \ldots, \tilde{C}_m)$, where $\tilde{C}_j = (\tilde{x}_{j_1}, \ldots, \tilde{x}_{j_s})$ is a noisy prediction of $C_j$. The third prediction model is our main focus.

**Definition 1.3** (Clause Advice). *In this model, for any noisy prediction $\tilde{C}_j$, $\tilde{x}_{j_k} = x_{j_k}^*$ with probability $(1 + \varepsilon)/2$ and $\tilde{x}_{j_k} = -x_{j_k}^*$ with probability $(1 - \varepsilon)/2$. Moreover, all $\tilde{C}_j$ are independent.*

We now briefly discuss the interplay between the three prediction models. The starting point of our discussion is the high-level algorithmic strategy employed in Cohen-Addad et al. (2024) and Ghoshal et al. (2025) for their main application of the MaxCut problem (given a graph, find a partition maximizing the number of edges cut). They use the label prediction model and a major part of their analysis boils down to confidently placing high-degree vertices on the correct side of the cut. Since high-degree vertices have many neighbors, one can 'boost' the success probability of placing high-degree nodes by looking at the aggregate assignment of their (large) neighborhoods (since intuitively in MaxCut, we want to separate high-degree vertices from *all* of their neighbors). However, in the context of Max-E3SAT, label advice doesn't seem to be powerful enough to carry out such a clean intuition as in MaxCut. To the best of our knowledge, label advice seems to give no improvement on the fundamental Max-E3SAT problem beyond the standard $7/8$ approximation.

The subset prediction model (Definition 1.2) is a significant strengthening of the label advice model, but we argue it makes the problem unnaturally easy. The subset prediction model gives us the exact assignment on a $\epsilon n$ sized subset of variables. Given this advice, a small modification of the original $7/8$ approximation immediately works: we simply randomly pick the assignment of all the other unobserved variables. The analysis is almost identical to the classic $7/8$ approximation: for a fixed clause, the variable that certifies that the clause evaluates to true is revealed with probability $\epsilon$. Otherwise, the random assignment satisfies the clause with probability $7/8$, giving that the overall probability of the clause being satisfied is $\geq \epsilon + (1 - \epsilon) \cdot 7/8 \geq 7/8 + \Theta(\epsilon)$ (indeed this observation was noted in a recent concurrent work of Attias et al. (2025). We discuss their paper more in Appendix A).

Note that this algorithm under the subset prediction model also does not require looking at the input at all! Furthermore, there is no element of uncertainty in the given advice (the bits that are revealed are always correct), meaning the algorithm design does not need to be robust against potentially untrustworthy advice. While we strongly believe there is ample room to introduce natural prediction models and the subset prediction advice certainly is natural in many other settings of CSPs (e.g. in the MaxCut example of Cohen-Addad et al. (2024)), we seek an alternate prediction model for the fundamental problem of Max-E3SAT for the above reasons (to exploit the input structure and incorporate noisy information).

This motivates our clause prediction model, given in Definition 1.3. Similar to the subset prediction model, our model is also a strengthening of the label advice model, but in a different manner. Our model intuitively gives noisy predictions *per constraint* rather than per variable. We believe this to be natural, as variables appear in many constraints can be thought of as 'important', and arguably a reliable ML predictor in practice should have more predictive power for important variables. For example, consider the setting of graphs. Many optimization problems on graphs can be modeled as CSPs where every edge represents a constraint (e.g. vertex cover or independent set). In that context, a per constraint prediction model such as ours gives

more information for higher degree nodes. Unlike subset constraints, we never know the exact right answer for any of the variables. That is, our prediction model allows for errors and uncertainty, and similar to the label advice model, every bit of information we receive is noisy, meaning our algorithm design must be robust against incorrect information. In the paragraph below, we discuss another advantage of our prediction model.

**Contributions Towards Question 2.** We now describe how our clause prediction model allows us to exploit structure of the underlying input to Max-E3SAT. At a high level, it allows us to use a 'high/low' degree decomposition design principal (a similar principal was used in Cohen-Addad et al. (2024) for Max-Cut) for Max-E3SAT. First we note that there exists an algorithm of Håstad (2000) (see Theorem 4.1 and the subsequent discussion) which obtains a better approximation factor for Max-3SAT for structured instances, where each variable appears in a bounded number of clauses. This inspires the following methodology. We first consider 'high degree' variables $x$ (i.e. variables that appear in a sufficiently large number of clauses), and use the majority of their prediction bits $\tilde{x}(C)$ for each clause $C$ that they appear in to decide how to set their assignments. Then intuitively, we want to simplify the given CSP by removing already satisfied clauses (in the case where they have a variable that is set to True) or shrinking their size (by removing variables that are set to False in the clause). Finally, we run an appropriate algorithm on this simplified instance which is more structured since only 'low-degree' variables remain. The idea is that on the one hand, for the high degree vertices, the majority vote is a very accurate prediction, and on the other hand, for CSPs with 'bounded degree', there exists a polynomial time algorithm with better approximation guarantees. This latter is the structure that we can finally exploit in our learning-augmented algorithm design! However, as detailed in the technical overview section (Section 1.2), care must be used in fully carrying out this intuition.

**Contributions Towards Question 3.** Using the aforementioned ideas (we give a more detailed technical overview in Section 1.2), we obtain our main theorem stated below.

**Theorem 1.4.** *There exists a polynomial-time algorithm in the Clause Advice model that given an unweighted formula of Max-E3SAT and advice $\tilde{C}$ finds an assignment with approximation factor at least $7/8 + \Theta(\varepsilon^2/\log(1/\varepsilon))$ in expectation, where $\varepsilon$ is the parameter of the clause prediction model.*

The theorem naturally generalizes to the weighted case; see Corollary D.1. We also remark that our main result has robustness, even if the predictions are arbitrarily corrupt, in the following two ways:

1. Our approximation factors consist of two terms: one coming from the classic bounds without predictions ($7/8$) and another term that represents the advantage of our method using clause predictions $\Theta(\epsilon^2/\log(1/\epsilon))$. We recover the original worst-case guarantee in the limit $\epsilon \to 0$, which represents the case when predictions that are pure random noise. However as $\epsilon$ increases, the quality of our prediction improves and our approximation factor correspondingly increases. As $\varepsilon \to 1$, the occurrence bound $B \to 0$, leading the algorithm to rely entirely on the majority vote and thus output the optimal assignment derived from the prediction.

2. We can always take multiple algorithm runs (either our algorithm initialized with different $\epsilon$ values or the classic $7/8$ approximation) and take the best solution at the end (the solution that satisfies the most number of clauses). This is because checking the quality of a given assignment is trivial (can be done in linear time), ensuring that e.g. we can always do as well as the classic $7/8$ approximation.

We complement our main result with the following lower bound, which gives a non-trivial limitation of our algorithm the clause prediction model. More generally, it applies to any algorithm which first simplifies the input formula for 'high degree' variables.

**Theorem 1.5.** *For all $\varepsilon$ sufficiently small, there exists an unweighted formula of Max-E3SAT, such that our main algorithm in the Clause Advice model cannot find an assignment with approximation factor larger than $7/8 + O(\sqrt{\varepsilon})$ in expectation, where $\varepsilon$ is the parameter of the model.*

Our last two results deal with hardness of the Max-E3SAT problem in the two other advice models discussed. Our hardness results rely on standard complexity theory assumptions (see Conjecture E.3 and Conjecture E.4), but do not fully settle the complexity of the problem in the two prediction models (e.g. we know from the discussion above that one can easily get $7/8 + \Theta(\epsilon)$ approximation in the subset prediction model). Nevertheless, we believe they are an important starting point in quantifying the power of the three models. We note that any hardness result for the Variable Subset Advice automatically applies to the Label Advice, since we can construct the Label Advice based on the Variable Subset Advice. The relationship between these two advice models is mentioned in (Ghoshal et al., 2025). Thus, we just need to study the setting of Variable Subset Advice.

**Theorem 1.6.** *Assume that the ETH and Linear Size PCP Conjecture hold. For every $\delta > 0$, there exists $\varepsilon_0 = \varepsilon_0(\delta)$ such that for every $\varepsilon \in (0, \varepsilon_0)$, there is no polynomial time algorithm for Max-E3SAT in the Variable Subset Advice model (or Label Advice model) with parameter $\varepsilon$ that given a $(1 - \delta)$-satisfiable formula returns a solution satisfying at least a $(7/8 + \delta)$-fraction of the clauses with probability at least $0.9$ over the random advice.*

We also focus on the Max-E3SAT(B) problem, a restricted (easier) variant where each variable occurs in at most $B$ clauses (see Definition 1.9). In particular, we are given $m$ clauses, each with 3 boolean variables in conjunctive normal form, where each variable occurs in at most $B$ clauses. Our goal is again to find an assignment of the boolean variables maximizing the number of satisfied clauses. We obtain the following hardness result, analogous to the classical result of Trevisan (2001) (see Section 1.3).

**Theorem 1.7.** *Assume that the ETH and Linear Size PCP Conjecture hold. For every $\delta > 0$, there exists $\varepsilon_0 = \varepsilon_0(\delta)$ such that for every $\varepsilon \in (0, \varepsilon_0)$, there is no polynomial time algorithm for Max-E3SAT(B) in the Variable Subset Advice model (or Label Advice model) with parameter $\varepsilon$ that given a $(1 - \delta)$-satisfiable formula returns a solution satisfying at least a $(7/8 + \Omega(1/\sqrt{B}) + \delta)$-fraction of the clauses with probability at least $0.9$ over the random advice.*

We remark that the $\delta$ we achieve in the theorems above is detailed in the full proofs (see Appendix E), and they follow from the PCP conjecture and is of the form $\delta = 1/\text{poly}(\log(1/\epsilon))$. We remark that this $\delta$ is much larger than any polynomial in $\epsilon$ as $\epsilon \to 0$ (e.g. $\delta \gg \epsilon^{0.0001}$).

**Organization** The paper is organized as follows. Subsection 1.3 establishes formal definitions of our problems and introduces some key notations. Section 2 presents our main algorithmic contribution for Max-E3SAT with clause advice. Appendix A discusses some additional related works and Appendix B formalizes some definitions for weighted variants of our problems. Appendix C presents the omitted proof of main Theorem 1.4. Appendix D contains some supplementary proofs deferred from Section 2. Appendix E provides the proofs of Theorem 1.6 and Theorem 1.7. Appendix F provides the proof of Theorem 1.5. Appendix G presents experimental results comparing our augmented algorithm against baselines for Max-E3SAT.

## 1.2 TECHNICAL OVERVIEW

Our high-level approach builds upon the framework introduced by Cohen-Addad et al. (2024), but we incorporate a novel and counterintuitive operation to achieve robust theoretical guarantees. The key idea in Cohen-Addad et al. (2024) is to reduce general (arbitrary-degree) instances of MaxCut to bounded-degree (denoted by $d$) instances using noisy vertex predictions. Specifically, they employ a single bit of prediction for every vertex, indicating which side of the optimal partition the vertex is on. However, the bit is only correct with probability $1/2 + \varepsilon$ for an error parameter $\varepsilon \in (0, 1/2)$. Through a technical argument, the authors in Cohen-Addad et al. (2024) are able to reduce arbitrary MaxCut instances with such predictions to the cases where $d \approx 1/\varepsilon^2$.

A natural extension of this approach is to consider analogous reductions for Max-E3SAT using either the Label Advice or Variable Subset Advice model (Definition 1.1 and Definition 1.2). Under these models, we provide hardness results (Theorem 1.6 and Theorem 1.7), showing that even with a single-bit prediction for every variable (indicating its value in an optimal assignment), Max-E3SAT is hard to approximate to within a factor of $7/8 + \delta$ and Max-E3SAT(B) is hard to approximate to within a factor of $7/8 + O(1/\sqrt{B}) + \delta$ for $\delta$ a specific function of $\varepsilon$. Our proof framework aligns with Ghoshal et al. (2025), employing two key reductions: from Max-3-Lin to Max-E3SAT, and Max-E3SAT to Max-E3SAT(B). While these reductions are established in prior work (Håstad, 2001; Trevisan, 2001), we demonstrate their compatibility with the prediction-augmented framework.

A more refined extension involves the Clause Advice model (Definition 1.3), which enables more accurate predictions for high-occurrence variables. Intuitively, under this model, the predicted values of frequently appearing variables align with their optimal assignment values with high probability. Leveraging this, we can reduce Max-E3SAT instances to Max-3SAT(B) instances, similar to Cohen-Addad et al. (2024). However, this reduction alone is insufficient for algorithmic improvement. While predictions for bounded-occurrence variables may reduce the size of some clauses, they do not inherently reduce the number of unknown clauses, limiting their utility. Moreover, the reduction in clause size is unpredictable, and shrinking the variable set alone is known to be inadequate for in designing algorithms for Max-E3SAT.

To address this limitation, we introduce a counterintuitive step: simultaneously constructing two bounded-occurrence instances-one *following* the predictions ($\phi_1$, Algorithm 1) and another *inverting* the predictions ($\phi_2$, Algorithm 1). The intuition is that when predictions assign $-1$ to excessive variables, inverting the predictions may yield a more effective assignment. By balancing trade-offs between different sub-algorithms on these two instances, we are able to handle edge cases and ensure robust performance guarantees. In particular, we mainly employ two classic algorithmic components as subroutines, namely MAX3SAT from Karloff & Zwick (1997); Zwick (2002) and MAX3SATB from Håstad (2000) (the required types of instances are followed to the names of algorithms). Our approach dynamically selects the best assignment based on approximation factor: When predictions affect only a few variables, either MAX3SAT($\phi_1$) or MAX3SAT($\phi_2$) yields an assignment with the strong approximation. However, when predictions affect many variables but indicate excessive FALSE values, assigning values against the predictions (via MAX3SATB($\phi_2$)) yields a great number of satisfied clauses. Otherwise, following the predictions (via MAX3SATB($\phi_1$)) performs well.

Additionally, we construct a specialized Max-E3SAT instance to show that the algorithms relying solely on Clause Advice model cannot achieve an approximation factor better than $7/8 + O(\sqrt{\varepsilon})$. In this instance, the approximation factor is dominated by the assignment of a quarter of the variables. By analyzing the predictions for these critical variables, we derive the upper bound of our main algorithm.

### 1.3 PRELIMINARIES AND NOTATION

Note that we defer the corresponding definitions of the weighted case to Appendix B.

**Definition 1.8** (Max-E3SAT). *In Max-E3SAT, we are given an formula that consists of $m$ clauses, where each clause contains exactly 3 boolean variables. The goal is to find an assignment of the boolean variables maximizing the number of satisfied clauses.*

**Definition 1.9** (Max-E3SAT(B)). *In Max-E3SAT(B), we are given an formula that consists of $m$ clauses, where each clause contains exactly 3 boolean variables and each variable occurs in at most $B$ clauses. The goal is to find an assignment of the boolean variables maximizing the number of satisfied clauses.*

Håstad (2001) proved that Max-E3SAT is hard to approximate within a factor of $7/8$ (assuming $P \neq NP$). And Trevisan (2001) proved that Max-E3SAT(B) is hard to approximate within a factor of $7/8 + O(1/\sqrt{B})$ (assuming $RP \neq NP$).

We denote by $\tilde{x}(C)$ the prediction of variable $x$ from the prediction of clause $C$. For (unweighted) Max-E3SAT, we define $occ(x)$ as the number of occurrences of variable $x$ in the different clauses.

We call a clause trivial if its satisfiability is fixed after assigning values to some variables in it, otherwise, we call a clause non-trivial. It is clear that a trivial clause is either satisfied or non-satisfied. If a non-trivial clause contains $k$ unassigned variables, we call it a non-trivial-$k$ clause. We use $OPT$ to denote the value of an optimal assignment for a given formula. For any variable $x$ and its advice $\tilde{x}(C_i)$ in clause $C_i$, the set $\{\tilde{x}(C_i)\}_{i \in S}$ contains advices with two possible opposite values, 1 and $-1$. We denote the $\mathrm{Majority}(\{\tilde{x}(C_i)\}_{i \in S})$ as follows: $\mathrm{Majority}(\{\tilde{x}(C_i)\}_{i \in S}) = 1$, if $\left( \sum_{i \in S} \tilde{x}(C_i) \right) \geq 0$, and $\mathrm{Majority}(\{\tilde{x}(C_i)\}_{i \in S}) = -1$, otherwise, where $S$ is a subset of $[n]$.

## 2 ALGORITHM FOR MAX-E3SAT WITH CLAUSE ADVICE

In this section, we present the algorithm in Theorem 1.4, deferring its proof to Appendix C. Our analysis of Algorithm 2 builds upon a powerful technique introduced by Håstad (2000). While the original work informally outlines this technique in the context of Max-E3SAT, it primarily states a more general theorem (Theorem 4.1 in Håstad (2000)) with a looser bound. The author notes that a tighter analysis exists for Max-E3SAT and provides a quick and half-page long sketch, but certain technical details—particularly the transition between linear and constant terms in the analysis of $|f_C|$ (specifically, the case when $|\alpha| = 1$ on Page 6)—are not fully elaborated. To ensure clarity and rigor, we present a complete and detailed proof of this technique (with proofs in the appendix, particularly of Lemma 2.4), filling in these gaps while preserving the original insight. Our case analysis of Håstad (2000)'s original bound differs in several analytical aspects of $|f_C|$, though the final conclusion remains consistent with Håstad (2000).

To demonstrate the technique, we need a few definitions, including the multilinear polynomial representation of Max-kSAT.

**Definition 2.1.** *Let $\phi$ be an unweighted formula of Max-kSAT and $C$ be a clause in $\phi$. Suppose that $C = x_1 \vee x_2 \vee \cdots \vee x_k$, where $x_1, x_2, \cdots, x_k$ are variables in $\phi$. The multilinear polynomial of $C$ is defined as*

$$f_C = 1 - \frac{(1 - x_1)(1 - x_2)\dots(1 - x_k)}{2^k} = \sum_{\alpha \subseteq [k]} p_\alpha x^\alpha, \tag{1}$$

*where $[k]$ is the set of integers $\{1, 2, \dots, k\}$ and $x^\alpha = \prod_{i \in \alpha} x_i$.*

If $x_i$ is assigned to True, we set $x_i = 1$ in $f_C$; otherwise, we set $x_i = -1$ in $f_C$. Then if $C$ is satisfied by an assignment of $x_1, x_2, \dots, x_k$, we have $f_C = 1$ under this assignment; otherwise, we have $f_C = 0$ under this assignment. Thus, the satisfiability of clause $C$ can be represented by $f_C$.

As an example, for a clause $C = (x \vee y \vee z)$ in an unweighted formula of Max-E3SAT, we have

$$f_C = 1 - \frac{(1 - x)(1 - y)(1 - z)}{8} = \frac{7 + x + y + z - xy - xz - yz + xyz}{8}. \tag{2}$$

**Definition 2.2.** *Let $\phi$ be a weighted formula of Max-kSAT with clauses $C_1, C_2, \cdots, C_l$ of total weight $m$ and $n$ variables $x_1, x_2 \dots, x_n$. The multilinear polynomial of $\phi$ is defined as*

$f_\phi = \sum_{j \in [l]} f_{C_j} = \sum_{\alpha \subseteq [n], |\alpha| \leq k} p_\alpha x^\alpha$, *and the sum of the absolute values of non-constant coefficients of $f_\phi$ is defined as $|f_\phi| = \sum_{\alpha \subseteq [n], 1 \leq |\alpha| \leq k} |p_\alpha|$, where $[n]$ is the set of integers $\{1, 2, \dots, n\}$ and $x^\alpha = \prod_{i \in \alpha} x_i$.*

The following lemma provides a tool for analyzing the approximation factor. We can see its detailed usage in how to recover the classic approximation algorithm of Max-E3SAT(B).

**Lemma 2.3.** *Let $\phi$ be a given weighted formula of Max-3SAT(B). Let the optimal assignment value of $\phi$ be $OPT$. Then $p_\emptyset + |f_\phi| \geq OPT$, where $p_\emptyset$ is the constant term of $f_\phi$.*

The technique utilizes $|f_\phi|$ to track the value of the assignment found. Briefly, for each step of this technique, by assigning values to some variables, we obtain a new instance $\psi$ such that $|f_\psi| \geq |f_\phi| - B$. Simultaneously, the constant term of $f_\psi$ increases by at least $1/8$ per step. The process repeats until all variables are assigned, ultimately yielding a performance guarantee for the final assignment. This is captured in the following lemma and corollary statements.

**Lemma 2.4** (Håstad (2000))**.** *Let $\phi$ be a given unweighted formula of Max-3SAT(B). There exists a polynomial-time algorithm that finds an assignment of value at least $p_\emptyset + |f_\phi|/(8B)$, where $p_\emptyset$ is the constant term of $f_\phi$.*

**Corollary 2.5** (Håstad (2000))**.** *Let $\phi$ be a given weighted formula of Max-3SAT(B). There exists a polynomial-time algorithm that finds an assignment of value at least $p_\emptyset + |f_\phi|/(8B)$, where $p_\emptyset$ is the constant term of $f_\phi$.*

*Proof.* In the weighted Max-3SAT(B) setting, $w(x) \leq B$ for each variable $x$, generalizing the unweighted constraint that $occ(x) \leq B$. And the new instance for each step of this technique is also a weighted formula of Max-3SAT(B). It is straightforward to verify that all remaining arguments apply identically to both the weighted and unweighted versions. $\square$

As a direct consequence, the technique allow us to recover the classic approximation for Max-E3SAT(B).

**Corollary 2.6.** *There exists a polynomial-time algorithm that given an weighted formula of Max-E3SAT(B) finds an assignment with approximation factor at least $7/8 + 1/(64B)$.*

We also extend the technique to general Max-3SAT(B) instances, yielding a computable approximation factor, thus making progress on Max-3SAT(B) that clasically lacks established approximation bounds. We note that $p_\emptyset$ is computable for any given $\phi$. For example, if $\phi$ consists of 3-size clauses of total weight $\varepsilon m$ and 2-size clauses of total weight $(1 - \varepsilon)m$, then $p_\emptyset = 7\varepsilon m/8 + 3(1 - \varepsilon)m/4 = (6 + \varepsilon)m/8$.

**Corollary 2.7.** *There exists a polynomial-time algorithm that given an weighted formula $\phi$ of Max-3SAT(B) with clauses of total weight $m$ finds an assignment with approximation factor at least $c + (1 - c)/(8B)$, where $p_\emptyset$ is the constant term of $f_\phi$ and $c = p_\emptyset/m$.*

We defer the proofs of the above theorems and lemmas to Appendix D. In addition, as we mentioned earlier, our main algorithm incorporates the following classic algorithmic component as a subroutine.

**Theorem 2.8** ((Karloff & Zwick, 1997; Zwick, 2002))**.** *There exists a polynomial-time algorithm that given an weighted formula of Max-3SAT finds an assignment with approximation factor at least $7/8$.*

### 2.1 THE MAIN ALGORITHM

Our algorithm incorporates several classic algorithmic components as subroutines. A key aspect of our algorithm is that it incorporates the counterintuitive step of inverting the predictions, which ultimately enables us to derive rigorous theoretical guarantees. We formalize the notations for these key components as follows: Denote by MAX3SAT: $\psi \to A$ the algorithm from Theorem 2.8, where $\psi$ is a weighted formula of Max-3SAT and $A$ is an assignment for the variables in $\psi$; Denote by MAX3SATB: $\psi \to A$ the algorithm from Lemma 2.4, where $\psi$ is an unweighted formula of Max-3SAT(B) and $A$ is an assignment for the variables in $\psi$.

To illustrate the execution of our main algorithm, we present a simple example. Consider a Max-3SAT formula $\phi$ that consists of clauses $A_i, B_i, C_i$ and a single clause $D$, where $A_i = (u \vee x_i \vee x_{i+1}), B_i =$

$(v \lor x_i \lor x_{i+1}), C_i = (w \lor \bar{x}_i \lor \bar{x}_{i+1}), D = (\bar{u} \lor \bar{v} \lor w), 1 \le i \le 100$. In this formula, the only high-occurrence variables are $u, v, w$. Assume that $m_u = m_v = -1$ and $m_w = 1$. In Algorithm 1, we assign $u = v = -1, w = 1$ in the first copy $\phi_1$, and $u = v = 1, w = -1$ in the second copy $\phi_2$. After the cleaning, in $\phi_1$, clauses $C_i$ and $D$ are satisfied, while clauses $A_i$ and $B_i$ simplify to $x_i \lor x_{i+1}$; in $\phi_2$, clauses $A_i$ and $B_i$ are satisfied, while clauses $C_i$ simplify to $\bar{x}_i \lor \bar{x}_{i+1}$ and clause $D$ is removed. Subsequently, in Algorithm 2, we solve these simplified formulas $\phi_1$ and $\phi_2$ using the aforementioned classic subroutines and return the assignment with best approximation factor.

**Proof Sketch of the Main Theorem 1.4** We begin by generating two complementary instances, $\phi_1$ and $\phi_2$, from the original instance $\phi$ by following and inverting the majority votes, respectively. Then we can conclude that after Algorithm 1, there are lower bounds on the expected optimal assignment values for both $\phi_1$ and $\phi_2$. Next, we leverage Lemma 2.3 to compute the constant terms $f_{\phi_1}$ and $f_{\phi_2}$. These constant terms are crucial for analyzing the performance of the four subroutines ($A_1$ to $A_4$) within Algorithm 2. Finally, we conduct a trade-off analysis to combine the performances before, eliminating the unknown parameters and yielding a rigorous lower bound.

---

**Algorithm 1** CLEANUP$(\phi, \tilde{C}_\phi, B)$

---

1: $\phi_1 \leftarrow \phi, \phi_2 \leftarrow \phi$.
2: **for** any variable $x$ with $occ(x) \ge B$ **do**
3:      $m_x \leftarrow$ Majority$(\{\tilde{x}(C_i)\}_{x \in C_i, i \in [m]})$.
4:      Assign $m_x$ to $x$ in $\phi_1$ and $-m_x$ to $x$ in $\phi_2$.
5: **end for**
6: **for** $i \in \{1, 2\}$ **do**
7:      **for** any trivial clause $C$ in $\phi_i$ **do**
8:          **if** $C$ is non-satisfied **then**
9:              Remove $C$ from $\phi_i$.
10:          **end if**
11:      **end for**
12: **end for**
13: **return** $(\phi_1, \phi_2)$

---

**Algorithm 2** MAXE3SAT-ADVICE$(\phi, \tilde{C}_\phi)$

---

1: $B \leftarrow 10 \log(1/\varepsilon)/\varepsilon^2$.
2: $(\phi_1, \phi_2) \leftarrow$ CLEANUP$(\phi, \tilde{C}_\phi, B)$.
3: $A_1 \leftarrow$ MAX3SATB$(\phi_1)$.
4: $A_2 \leftarrow$ MAX3SATB$(\phi_2)$.
5: $A_3 \leftarrow$ MAX3SAT$(\phi_1)$.
6: $A_4 \leftarrow$ MAX3SAT$(\phi_2)$.
7: **return** $A$ with best approximation factor among $\{A_1, A_2, A_3, A_4\}$.

---

## 3   CONCLUSION AND OPEN PROBLEMS

We propose a natural clause prediction model for Max-E3SAT that enables us to design an algorithm to go beyond the classical worst-case lower bounds of $7/8$, achieving an approximation ratio of $7/8 + \Theta(\varepsilon^2/\log(1/\varepsilon))$. Our algorithm integrates several classical algorithms as subroutines, combined with a counterintuitive step of inverting predictions. For hardness, we show that Max-E3SAT is hard to approximate to within a factor of $7/8 + \delta$ and Max-E3SAT with bounded-occurrence $B$ is hard to approximate to within a factor of $7/8 + O(1/\sqrt{B}) + \delta$ for $\delta$ a specific function of $\varepsilon$. We further identify the following natural open questions following our work, which we believe are interesting directions in incorporating advice in fundamnetal optimization problems:

1. What is the best approximation algorithm that we can get under the label advice prediction model (Definition 1.2)? Our Theorems (1.6 and 1.7) give a lower bound, but we have no upper bound results. We conjecture that one cannot improve upon the $7/8$ approximation factor for sufficiently small constant $\epsilon$.

2. Similarly, what is the right polynomial dependence on the advantage one can get beyond $7/8$ using clause predictions? There is still a polynomial gap between our upper and lower bounds of Theorem 1.4 and

1.5. Furthermore, proving a lower bound similar to our Theorem 1.5, but which holds for all possible algorithms (e.g. based on a hardness assumption) is an intersting future direction.

3. More broadly, what are other CSPs that can benefit from the clause prediction model that we introduce? One candidate would be Max-EkSAT for other values of $k > 3$, which is the natural extension of maximizing the number of satisfiable clauses where every clause has exactly $k$ variables. One bottleneck here is that one would need to first make fundamental progress on approximation algorithms themselves (without advice). This is because we are not aware of a similar result as in Theorem (Håstad, 2000) giving a better approximation factor for bounded instances for general $k$. Even more surprisingly, while Max-EkSAT classically admits a trivial $1 - 2^{-k}$ approximation by picking a random assignment, this is not true for the version of the problem where clauses can have different number of variables (up to $k$). Note that for the $k = 3$ case, it was shown in Karloff & Zwick (1997); Zwick (2002) (see Theorem 2.8) that one can obtain a $7/8$ approximation in polynomial time if the clauses can have a different number of variables, up to 3, via a complicated SDP and computer assisted proof.

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

## A  OTHER RELATED WORKS

A recent independent and concurrent work by (Attias et al., 2025) also studies Max-3SAT with advice. They propose an approximation algorithm achieving a factor of $7/8 + \Omega(\varepsilon)$ in the Variable Subset Advice model. However, their results are confined to this model and cannot be generalized to the Label Advice model. Their work and ours represent two parallel advancements in generalizing the Label Advice model: whereas (Attias et al., 2025) enhance robustness and reliability via Variable Subset Advice, our work introduces the Clause Advice model, which amplifies prediction accuracy while accommodating uncertainty. Their improvement aligns with findings by (Cohen-Addad et al., 2024), who show that Variable Subset Advice model can admit better algorithms than Label Advice model. Notably, the algorithmic proof of (Attias et al., 2025) is very short (we outline it in our introduction, see Section 1), which can demonstrate how uncertainty inherently limits the design of stronger algorithms (without uncertainty, designing an algorithm is easy)—further highlighting the value of our work.

Lastly, we remark that while our work is the first to introduce clause predictions (a prediction for each constraint) for augmenting CSPs, we remark that similar 'per constraint' predictions have also been used in other learning-augmented optimization problems (unrelated to CSPS), e.g. (Silwal et al., 2023) for correlation clustering and (Bateni et al., 2024) for metric clustering.

## B  OMITTED PRELIMINARIES

**Definition B.1** (Weighted Max-E3SAT). *In weighted Max-E3SAT, we are given an formula that consists of clauses with total weight $m$, where each clause contains exactly 3 boolean variables. The goal is to find an assignment of the boolean variables maximizing the total weight of satisfied clauses.*

For weighted Max-E3SAT, we define $w(C)$ as the weight of clause $C$ and define $w(x)$ as the total weight of clauses that contains $x$. Note that $w(x) = \sum_{x \in C} w(C)$.

Note that $v$ and $-v$ are denoted as two possible opposite values of advices in $\{\tilde{x}(C_i)\}_{i \in S}$. We define WMajority($\{w_{C_i}, \tilde{x}(C_i)\}_{i \in S}$) as follows: WMajority($\{w_{C_i}, \tilde{x}(C_i)\}_{i \in S}$) = $v$, if $\left( \sum_{i \in S} w_{C_i} \tilde{x}(C_i) \right) \cdot v \geq 0$, and WMajority($\{w_{C_i}, \tilde{x}(C_i)\}_{i \in S}$) = $-v$, otherwise, where $S$ is a subset of $[n]$.

As in Section 2.1, denote by WMAX3SATB: $\psi \rightarrow A$ the algorithm from Corollary 2.5, where $\psi$ is a weighted formula of Max-3SAT(B) and $A$ is an assignment for the variables in $\psi$;

As in Section 2, we can also consider the weighted version of mutilinear polynomial of clause as follows. Notably, the unweighted and weighted versions share an identical multilinear polynomial representation, which enables us to generalize any unweighted algorithmic result to its weighted variant.

**Definition B.2.** *Let $\phi$ be an weighted formula of Max-kSAT and $C$ be a clause of weight $w_C$ in $\phi$. Suppose that $C = x_1 \vee x_2 \vee \cdots \vee x_k$, where $x_1, x_2, \cdots, x_k$ are variables in $\phi$. The multilinear polynomial of $C$ is defined as*

$$f_C = w_C \left( 1 - \frac{(1-x_1)(1-x_2)\dots(1-x_k)}{2^k} \right) = \sum_{\alpha \subseteq [k]} p_\alpha x^\alpha, \tag{3}$$

*where $[k]$ is the set of integers $\{1, 2, \dots, k\}$ and $x^\alpha = \prod_{i \in \alpha} x_i$.*

Building upon the definition of the multilinear polynomial of clause, we now introduce the definition of the multilinear polynomial of formula, which serves as the foundation for our subsequent algorithmic analysis.

## C  OMITTED PROOF OF MAIN THEOREM 1.4

*Proof of Main Theorem 1.4.* Let $\phi$ be the given formula with $m$ clauses $C_1, C_2 \ldots, C_m$ and $n$ variables. Note that both MAX3SATB and MAX3SAT are polynomial-time algorithms. So Algorithm 2 is also a polynomial-time algorithm.

We claim that the return $\phi_1$ of Algorithm 1 has the expected optimal assignment value at least $(1-\varepsilon^5) \cdot OPT$.

Consider any variable $x$ with $occ(x) \geq B$. W.l.o.g., let $C_i$ be the clause such that $x \in C_i$, where $1 \leq i \leq occ(x)$. Let $X_i$ be the random variable such that $X_i = 1$ when $\tilde{x}(C_i) = x^*$ and $X_i = -1$ when $\tilde{x}(C_i) = -x^*$. By the definition of Clause Advice, we have $\Pr[X_i = 1] = (1+\varepsilon)/2$ and $\Pr[X_i = -1] = (1 - \varepsilon)/2$. Let $X = \sum_{i=1}^{occ(x)} X_i$. Then $\mathbb{E}[X] = occ(x) \cdot \varepsilon \geq B\varepsilon$. Here we have $X \geq 0$ if and only if $\mathrm{Majority}(\{\tilde{x}(C_i)\}_{x \in C_i, i \in [m]}) = x^*$. By Hoeffding's inequality,

$$\Pr[X \leq 0] \leq \exp(-occ(x) \cdot \varepsilon^2/2) \leq \exp(-B\varepsilon^2/2) = \varepsilon^5. \tag{4}$$

Then the probability that $\mathrm{Majority}(\{\tilde{x}(C_i)\}_{x \in C_i, i \in [m]}) = -x^*$ is at most $\varepsilon^5$.

Select arbitrarily a satisfiable clause $C$ for the fixed optimal assignment $x^*$. Since $C$ is satisfiable, there must be at least one variable $x_C$ in $C$ such that $x_C^* = 1$. If $x_C$ is not a high-occurrence variable, $C$ is still satisfiable since we do not assign any value to $x_C$. If $x_C$ is a high-occurrence variable, $C$ is still satisfiable when $\mathrm{Majority}(\{\tilde{x_C}(C_i)\}_{x_C \in C_i, i \in [m]}) = x_C^*$. We denote by $S(x_C)$ the set of satisfiable clauses that contains $x_C$ and by $E(x_C)$ the event that $\mathrm{Majority}(\{\tilde{x_C}(C_i)\}_{x_C \in C_i, i \in [m]}) = x_C^*$. For the same reason, any clause in $S(x_C)$ is still satisfiable when $E(x_C)$ happens. Once we find such $S(x_C)$, we can remove $S(x_C)$ from the instance and keep looking for a new $x_C$ and corresponding $S(x_C)$. By this way, we can find some disjoint sets $S(x_j)_{1 \leq j \leq s}$ such that any clause in $S(x_j)$ is still satisfiable when $E(x_j)$ happens. Since the events $E(x_j)_{1 \leq j \leq s}$ are independent (note that all $\tilde{C}_i$ are independent), we can conclude that the expected optimal assignment value is at least $\sum_{j=1}^{s} \Pr[E(x_C)] \cdot |S(x_j)| \geq (1-\varepsilon^5) \cdot \sum_{j=1}^{s} |S(x_j)| = (1-\varepsilon^5) \cdot OPT$.

Thus, the return $\phi_1$ of Algorithm 1 has the expected optimal assignment value at least $(1 - \varepsilon^5) \cdot OPT$.

Let $m' = (1 - \varepsilon^5) \cdot OPT$. In Algorithm 1, we remove any non-satisfied clause from the resulting clauses $\phi_1$ and $\phi_2$. Note that non-trivial-3 clause has no assigned variables. Suppose that the final output of $\phi_i$ consists of $\alpha_i$ non-trivial-1 clauses, $\beta_i$ non-trivial-2 clauses, $\gamma_i$ satisfied clauses and $\zeta$ non-trivial-3 clauses, where $i \in \{1, 2\}$. Since the assignments are completely inverse in $\phi_1$ and $\phi_2$, we have $\gamma_1 \geq \alpha_2 + \beta_2$ and $\gamma_2 \geq \alpha_1 + \beta_1$. Then the expected optimal assignment value for $\zeta$ non-trivial-3 clauses is at least $m' - \alpha_1 - \beta_1 - \gamma_1$. Thus, the final output of $\phi_2$ has the expected optimal assignment value at least $m' - \alpha_1 - \beta_1 - \gamma_1 + \gamma_2 \geq m' - \gamma_1$.

In Algorithm 2, we execute MAX3SATB($\phi_1$) and MAX3SATB($\phi_2$). Let $f_{\phi_1} = \sum_{\alpha \subseteq [n], |\alpha| \leq 3} p_\alpha x^\alpha$ and $f_{\phi_2} = \sum_{\alpha \subseteq [n], |\alpha| \leq 3} q_\alpha x^\alpha$. By Lemma 2.4, MAX3SATB($\phi_1$) outputs an assignment of value at least $p_\emptyset + |f_{\phi_1}|/(8B)$ and MAX3SATB($\phi_2$) outputs an assignment of value at least $q_\emptyset + |f_{\phi_2}|/(8B)$. By Lemma 2.3, $p_\emptyset + |f_{\phi_1}| \geq m'$ and $q_\emptyset + |f_{\phi_2}| \geq m' - \gamma_1$ in expectation.

Let us analyze $p_\emptyset$ and $q_\emptyset$. According to the definition of $f_{\phi_1}$, any satisfied clause contributes 1 and any non-trivial-$k$ clause contributes $1 - 2^{-k}$, where $k \in \{1, 2, 3\}$. Since $m' \leq \alpha_1 + \beta_1 + \gamma_1 + \zeta$, we have $p_\emptyset \geq 7m'/8 - 3\alpha_1/8 - \beta_1/8 + \gamma_1/8$. Likewise, $q_\emptyset \geq 7m'/8 - 7\gamma_1/8 - 3\alpha_2/8 - \beta_2/8 + \gamma_2/8$.

To simplify the following calculations, we define the weighted expected approximation factor as the expected approximation factor times $OPT/m' = 1/(1 - \varepsilon^5)$. Thus, the weighted expected approximation factor of

$A_1$ is at least

$$\frac{p_\emptyset + |f_{\phi_1}|/(8B)}{m'} \geq \frac{7}{8} + \frac{(-3\alpha_1 - \beta_1 + \gamma_1)(8B-1)}{64Bm'} + \frac{1}{64B}$$
$$\geq \frac{7}{8} + \frac{(-3\alpha_1 - 3\beta_1 + \gamma_1)(8B-1)}{64Bm'} + \frac{1}{64B}$$

and the weighted expected approximation factor of $A_2$ is at least

$$\frac{q_\emptyset + |f_{\phi_2}|/(8B)}{m'} \geq \frac{7}{8} + \frac{(-3\alpha_2 - \beta_2 + \gamma_2)(8B-1) - \gamma_1(56B+1)}{64Bm'} + \frac{1}{64B}$$
$$\geq \frac{7}{8} + \frac{(-3\alpha_2 - 3\beta_2 + \alpha_1 + \beta_1)(8B-1) - \gamma_1(56B+1)}{64Bm'} + \frac{1}{64B}$$
$$\geq \frac{7}{8} + \frac{(-3\gamma_1 + \alpha_1 + \beta_1)(8B-1) - \gamma_1(56B+1)}{64Bm'} + \frac{1}{64B}$$
$$\geq \frac{7}{8} + \frac{(-11\gamma_1 + \alpha_1 + \beta_1)(8B-1)}{64Bm'} + \frac{1}{64B},$$

where we use $8B \geq 9$ in the last inequality.

On the other hand, by Theorem 2.8, the weighted expected approximation factor of $A_3$ is at least

$$\frac{\gamma_1 + 7(m' - \gamma_1)/8}{m'} = \frac{7}{8} + \frac{\gamma_1}{8m'} \geq \frac{7}{8} + \frac{\gamma_1(8B-1)}{64Bm'}$$

and the weighted expected approximation factor of $A_4$ is at least

$$\frac{\gamma_2 + 7(m' - \gamma_1 - \gamma_2)/8}{m'} = \frac{7}{8} + \frac{-7\gamma_1 + \gamma_2}{8m'} \geq \frac{7}{8} + \frac{(-7\gamma_1 + \alpha_1 + \beta_1)(8B-1)}{64Bm'}.$$

Let $X = \alpha_1 + \beta_1$ and $Y = \gamma_1$. Let the weighted expected approximation factor of $A$ be $\mathcal{M}$. Then

$$\mathcal{M} \geq \frac{7}{8} + \frac{\max\{\rho_1, \rho_2, \rho_3, \rho_4\}(8B-1) + m'}{64Bm'},$$

where $\rho_1 = Y - 3X$, $\rho_2 = X - 11Y$, $\rho_3 = Y - m'/(8B-1)$ and $\rho_4 = (X - 7Y) - m'/(8B-1)$.

1. when $X < Y/3$ or $X \geq 11Y$, $\mathcal{M} \geq 7/8 + 1/(64B)$;

2. when $8Y > X \geq Y/3$, $\mathcal{M} \geq 7/8 + 1/(576B)$;

3. when $11Y > X \geq 8Y$, $\mathcal{M} \geq 7/8 + 1/(256B)$;

The tight boundary for $\mathcal{M} \geq 7/8 + 1/(576B)$ is that

$$X = \frac{m'}{3(8B-1)} \text{ and } Y = \frac{m'}{9(8B-1)}.$$

So the expected approximation factor of $A$ is at least

$$(7/8 + 1/(576B)) \cdot (1 - \varepsilon^5) = 7/8 + \Theta(\varepsilon^2/\log(1/\varepsilon)) \tag{5}$$

Therefore, Algorithm 2 finds an assignment with approximation factor at least $7/8 + \Theta(\varepsilon^2/\log(1/\varepsilon))$ in expectation. $\qquad \square$

## D  OMITTED PROOFS OF SECTION 2

In this section, we supplement the proofs omitted in section 2, starting by proving that Lemma 2.3.

*Proof of Lemma 2.3.* Note that $p_\emptyset \geq 0$. Since $OPT = f_\phi(x_1^*, x_2^*, \ldots, x_n^*)$ where $(x_1^*, x_2^*, \ldots, x_n^*)$ is the optimal assignment of $\phi$ and $x_i^* \in \{-1, 1\}$, we have $OPT \leq \sum_{\alpha \subseteq [n], |\alpha| \leq k} |p_\alpha| = p_\emptyset + |f_\phi|$. ☐

As stated at the beginning of section 2, we provide a complete and detailed proof of the technique introduced by (Håstad, 2000). We construct a new Max-3SAT(B) formula $\psi$ from an original formula $\phi$ such that $|f_\psi| < |f_\phi|$, where $|f_\cdot|$ is the key measurement defined in Definition 2.2. Following the methodology of (Håstad, 2000), we perform a rigorous case analysis to establish a lower bound on the reduction $|f_\phi| - |f_\psi|$ at each transformation step. Combining this bound with Lemma 2.3 yields the final conclusion.

*Proof of Lemma 2.4.* Suppose that $\phi$ consists of $m$ clauses and $n$ variables $x_1, x_2 \ldots, x_n$. Note that $f_\phi = \sum_{\alpha \subseteq [n], |\alpha| \leq 3} p_\alpha x^\alpha$ and $|f_\phi| = \sum_{\alpha \subseteq [n], 1 \leq |\alpha| \leq 3} |p_\alpha|$. Let $\beta$ be the minimal set such that $p_\beta \neq 0$ and $p_\gamma = 0$ for any $\emptyset \neq \gamma \subset \beta$, where the minimality refers to the size of the set. Such $\beta$ exists when $f_\phi$ is non-trivial. Find an assignment in $\{-1, 1\}^\beta$ to the variables in $\beta$ such that $p_\beta x^\beta = |p_\beta|$. After this assignment, we can get a new formula $\psi$. We similarly define $f_\psi = \sum_{\alpha \subseteq [n], |\alpha| \leq 3} q_\alpha x^\alpha$ and $|f_\psi| = \sum_{\alpha \subseteq [n], 1 \leq |\alpha| \leq 3} |q_\alpha|$.

Leveraging the minimality of $\beta$, we can analyze $|f_\phi| - |f_\psi|$ in the following cases:

1. Suppose that $|\beta| = 1$. W.l.o.g, let $x^\beta = x_1$. Consider any clause $C$ that contains $x_1$. Define the multilinear polynomial of $C$ as $f_C$. In $f_C$, any non-linear term with $x_1$ may lead to the cancellation in $f_\psi$ and the linear term of $x_1$ can become the part of $q_\emptyset$.

   (a) When $C$ is 1-size, $f_C = 1/2 + x_1/2$. The corresponding component of $|f_\phi| - |f_\psi|$ is bounded by $1/2$.

   (b) When $C$ is 2-size, $f_C = 3/4 + x_1/4 + y/4 - x_1 y/4$ where $y$ is another variable in $C$. The corresponding component of $|f_\phi| - |f_\psi|$ is bounded by $1/4 + 2 \cdot 1/4 = 3/4$.

   (c) When $C$ is 3-size, $f_C = 7/8 + x_1/8 + y/8 + z/8 - x_1 y/8 - x_1 z/8 - yz/8 + x_1 yz/8$ where $y$ and $z$ are other two variables in $C$. The corresponding component of $|f_\phi| - |f_\psi|$ is bounded by $1/8 + 2 \cdot 1/8 + 2 \cdot 1/8 + 2 \cdot 1/8 = 7/8$.

   Since $x_1$ appears in at most $B$ clauses, $|f_\phi| - |f_\psi|$ can be bounded by $7B/8$.

2. Suppose that $|\beta| = 2$. Let $x^\beta = x_1 \cdot x_2$. Consider any clause $C$ that contains $a$ or $b$. Define the multilinear polynomial of $C$ as $f_C$. By the minimality of $\beta$, we know that any coefficient of linear term in $f_\phi$ is 0. So when we analyze $f_C$, we can directly eliminate the linear term. This elimination makes our analysis simple and does not affect our analysis about the corresponding component of $|f_\phi| - |f_\psi|$. We ignore the trivial case that $C$ is 1-size in the following analysis.

   (a) When $C$ is 2-size and $x_1 \in C, x_2 \notin C$ (w.l.o.g), $f_C = 3/4 - x_1 y/4$ where $y$ is another variable in $C$. The corresponding component of $|f_\phi| - |f_\psi|$ is bounded by $2 \cdot 1/4 = 1/2$.

   (b) When $C$ is 2-size and $x_1, x_2 \in C$, $f_C = 3/4 - x_1 x_2/4$. The corresponding component of $|f_\phi| - |f_\psi|$ is bounded by $1/4$.

   (c) When $C$ is 3-size and $x_1 \in C, x_2 \notin C$ (w.l.o.g), $f_C = 7/8 - x_1 y/8 - x_1 z/8 - yz/8 + x_1 yz/8$ where $y$ and $z$ are other two variables in $C$. The corresponding component of $|f_\phi| - |f_\psi|$ is bounded by $1/8 + 1/8 + 2 \cdot 1/8 = 1/2$, since the boundary case satisfies the conditions that the coefficient of term $y$ or $z$ is 0 in $f_\psi$.

(d) When $C$ is 3-size and $x_1, x_2 \in C$, $f_C = 7/8 - x_1 x_2/8 - x_1 y/8 - x_2 y/8 + x_1 x_2 y/8$ where $y$ is another variable in $C$. The corresponding component of $|f_\phi| - |f_\psi|$ is bounded by $1/8 + 1/8 + 1/8 + 1/8 = 1/2$, since the boundary case is that the coefficient of term $y$ is 0 in $f_\psi$.

Since $x_1$ or $x_2$ appears in at most $B$ clauses, $|f_\phi| - |f_\psi|$ can be bounded by $B/2 + B/2 = B$.

3. Suppose that $|\beta| = 3$. Let $x^\beta = x_1 \cdot x_2 \cdot x_3$. Consider any clause $C$ that contains $a$ or $b$. Define the multilinear polynomial of $C$ as $f_C$. By the minimality of $\beta$, we know that any coefficient of linear term or quadratic term in $f_\phi$ is 0. For the similar reason as above, we will eliminate the linear term and quadratic term in $f_C$. We ignore the trivial case that $C$ is 1-size or 2-size in the following analysis.

   (a) When $x_1 \in C, x_2, x_3 \notin C$ (w.l.o.g), $f_C = 7/8 + x_1 y z/8$ where $y$ and $z$ are other two variables in $C$. The corresponding component of $|f_\phi| - |f_\psi|$ is bounded by $2 \cdot 1/8 = 1/4$.

   (b) When $x_1, x_2 \in C, x_3 \notin C$ (w.l.o.g), $f_C = 7/8 + x_1 x_2 y/8$ where $y$ is another variable in $C$. The corresponding component of $|f_\phi| - |f_\psi|$ is bounded by $2 \cdot 1/8 = 1/4$.

   (c) When $x_1, x_2, x_3 \in C$, $f_C = 7/8 + x_1 x_2 x_3/8$. The corresponding component of $|f_\phi| - |f_\psi|$ is bounded by $1/8$.

   Since $x_1, x_2$ or $x_3$ appears in at most $B$ clauses, $|f_\phi| - |f_\psi|$ can be bounded by $B/4 + B/4 + B/4 = 3B/4$.

From the above analysis, we can know that $|f_\phi| - |f_\psi| \leq B$. Remind that $q_\emptyset - p_\emptyset = |p_\beta| \geq 1/8$. We note that the formula $\psi$ is also an unweighted formula of Max-3SAT (B). That means we can repeatedly find some assignment to get a new formula until the new formula $\phi^*$ has a trivial multilinear polynomial $f_{\phi^*} = p_\emptyset^*$. Then $p_\emptyset^* \geq p_\emptyset + |f_\phi|/(8B)$. □

We recover the celebrated $7/8 + 1/(64B)$-approximation algorithm for Max-E3SAT(B) by combining existing results. Specifically, we apply Corollary 2.5 to find the assignment, and derive the approximation guarantee utilizing Lemma 2.3 and the fact that the multilinear polynomial of any Max-E3SAT(B) formula has the fixed constant term $7m/8$.

*Proof of Corollary 2.6.* Let $\phi$ be the given formula with clauses of total weight $m$ and $n$ variables $x_1, x_2 \ldots, x_n$. Note that $f_\phi = \sum_{\alpha \subseteq [n], |\alpha| \leq 3} p_\alpha x^\alpha$ and $|f_\phi| = \sum_{\alpha \subseteq [n], 1 \leq |\alpha| \leq 3} |p_\alpha|$. Since $\phi$ is an weighted formula for Max-E3SAT(B), we have $p_\emptyset = 7m/8$. By Corollary 2.5, there exists a polynomial-time algorithm that finds an assignment of value at least $p_\emptyset + |f_\phi|/(8B)$. By Lemma 2.3, $p_\emptyset + |f_\phi| \geq OPT$. And it is trivial that $m \geq OPT$. Then the performance ratio is at least

$$\frac{p_\emptyset + |f_\phi|/(8B)}{\min\{m, p_\emptyset + |f_\phi|\}} = \frac{7m/8 + |f_\phi|/(8B)}{\min\{m, 7m/8 + |f_\phi|\}}$$

When $|f_\phi| \geq m/8$, we have

$$\frac{7m/8 + |f_\phi|/(8B)}{\min\{m, 7m/8 + |f_\phi|\}} = \frac{7m/8 + |f_\phi|/(8B)}{m} \geq 7/8 + 1/(64B).$$

When $|f_\phi| \leq m/8$, we have

$$\frac{7m/8 + |f_\phi|/(8B)}{\min\{m, 7m/8 + |f_\phi|\}} = \frac{7m/8 + |f_\phi|/(8B)}{7m/8 + |f_\phi|} \geq 7/8 + 1/(64B).$$

The last step holds when $8B \geq 1$. For the sake of simplicity, we let $a = m/8$, $b = |f_\phi|$ and $k = 8B$. When $a \geq b$ and $k \geq 1$, we have

$$
\begin{aligned}
\frac{7a + b/k}{7a + b} &= 7/8 + \frac{7a/8 - 7b/8 + b/k}{7a + b} \\
&= 7/8 + 1/(8k) + \frac{7a/8 - 7b/8 - 7a/(8k) + 7b/(8k)}{7a + b} \\
&= 7/8 + 1/(8k) + \frac{(7/8) \cdot (a - b) \cdot (1 - 1/k)}{7a + b} \\
&\geq 7/8 + 1/(8k).
\end{aligned}
$$

Therefore, we can find an assignment with approximation factor at least $7/8 + 1/(64B)$. $\square$

The proof of Corollary 2.7 closely parallels that of Corollary 2.6. However, since the value of $p_\emptyset$ is not fixed for a general Max-3SAT(B) instance $\phi$, we treat it as a parameter.

*Proof of Corollary 2.7.* We present only the modified portion of the proof of Corollary 2.6, where we parameterize $p_\emptyset$.

The performance ratio is at least

$$
\frac{p_\emptyset + |f_\phi|/(8B)}{\min\{m, p_\emptyset + |f_\phi|\}} = \frac{cm + |f_\phi|/(8B)}{\min\{m, cm + |f_\phi|\}}
$$

When $|f_\phi| \geq (1 - c)m$, we have

$$
\frac{cm + |f_\phi|/(8B)}{\min\{m, cm + |f_\phi|\}} = \frac{cm + |f_\phi|/(8B)}{m} \geq c + (1 - c)/(8B).
$$

When $|f_\phi| \leq (1 - c)m$, we have

$$
\frac{cm + |f_\phi|/(8B)}{\min\{m, cm + |f_\phi|\}} = \frac{cm + |f_\phi|/(8B)}{cm + |f_\phi|} \geq c + (1 - c)/(8B).
$$

The last step holds when $8B \geq 1$. For the sake of simplicity, we let $a = (1 - c)m$, $b = |f_\phi|$ and $k = 8B$. When $a \geq b$ and $k \geq 1$, we have

$$
\begin{aligned}
\frac{ac/(1 - c) + b/k}{ac/(1 - c) + b} &= c + \frac{ac - bc + b/k}{ac/(1 - c) + b} \\
&= c + (1 - c)/k + \frac{ac - bc - ac/k + bc/k}{ac/(1 - c) + b} \\
&= c + (1 - c)/k + \frac{c \cdot (a - b) \cdot (1 - 1/k)}{ac/(1 - c) + b} \\
&\geq c + (1 - c)/k.
\end{aligned}
$$

Therefore, we can find an assignment with approximation factor at least $c + (1 - c)/(8B)$. $\square$

We further extend our analysis to the weighted variant of Theorem 1.4, which we formally present as Corollary D.1. The subroutine WMAX3SATB is defined in Appendix B.

**Corollary D.1.** *There exists a polynomial-time algorithm in the Clause Advice model that given an weighted formula of Max-E3SAT and advice $\tilde{C}$ finds an assignment with approximation factor at least $7/8 + \Theta(\varepsilon^2 / \log(1/\varepsilon))$ in expectation, where $\varepsilon$ is the parameter of the model.*

*Proof.* We adapt the algorithm for (unweighted) Max-E3SAT to the weighted case by making straightforward substitutions. In particular, we

- substitute $occ(x) \geq B$ with $w(x) \geq B$,

- substitute $\mathrm{Majority}(\{\tilde{x}(C_i)\}_{x \in C_i, i \in [m]})$ with $\mathrm{WMajority}(\{w_{C_i}, \tilde{x}(C_i)\}_{x \in C_i, i \in [m]})$, and

- substitute MAX3SATB from Lemma 2.4 with WMAX3SATB from Corollary 2.5.

These substitutions only require us to modify the relevant statements to preserve the original proof methodology.

---

**Algorithm 3** WEIGHTED-CLEANUP$(\phi, \tilde{C}_\phi, B)$

1: $\phi_1 \leftarrow \phi$, $\phi_2 \leftarrow \phi$.
2: **for** any variable $x$ with $w(x) \geq B$ **do**
3:      $m_x \leftarrow \mathrm{WMajority}(\{w_{C_i}, \tilde{x}(C_i)\}_{x \in C_i, i \in [m]})$.
4:      Assign $m_x$ to $x$ in $\phi_1$ and $-m_x$ to $x$ in $\phi_2$.
5: **end for**
6: **for** $i \in \{1, 2\}$ **do**
7:      **for** any trivial clause $C$ in $\phi_i$ **do**
8:          **if** $C$ is non-satisfied **then**
9:              Remove $C$ from $\phi_i$.
10:          **end if**
11:      **end for**
12: **end for**
13: **return** $(\phi_1, \phi_2)$

---

**Algorithm 4** WEIGHTED-MAXE3SAT-ADVICE$(\phi, \tilde{C}_\phi)$

1: $B \leftarrow 10 \log(1/\varepsilon)/\varepsilon^2$.
2: $(\phi_1, \phi_2) \leftarrow$ CLEANUP$(\phi, \tilde{C}_\phi, B)$.
3: $A_1 \leftarrow$ WMAX3SATB$(\phi_1)$.
4: $A_2 \leftarrow$ WMAX3SATB$(\phi_2)$.
5: $A_3 \leftarrow$ MAX3SAT$(\phi_1)$.
6: $A_4 \leftarrow$ MAX3SAT$(\phi_2)$.
7: **return** $A$ with best approximation factor among $\{A_1, A_2, A_3, A_4\}$.

---

In the weighted Max-3SAT(B) setting, $w(x) \leq B$ for each variable $x$, generalizing the unweighted constraint where $occ(x) \leq B$. We can still claim that the final output of $\phi_1$ has the expected optimal assignment value at least $(1 - \varepsilon^5) \cdot OPT$. Here we consider any variable $x$ with $w(x) \geq B$. Let $C_i$ be the clause such that $x \in C_i$, where $1 \leq i \leq k$. We can know that $\sum_{i=1}^{k} w_{C_i} = w(x)$. Similarly, we construct some random variables $X_i$ such that $X_i = 1$ when $\tilde{x}(C_i) = x^*$ and $X_i = -1$ when $\tilde{x}(C_i) = -x^*$, where $1 \leq i \leq k$. Let $X = \sum_{i=1}^{k} w_{C_i} \cdot X_i$. Then we have $\mathbb{E}[X] \geq B\varepsilon$ and $X \geq 0$ if and only if $\mathrm{WMajority}(\{w_{C_i}, \tilde{x}(C_i)\}_{x \in C_i, i \in [m]}) = x^*$. Also, by Hoeffding's inequality, we can know that $\Pr[X \leq 0] = \varepsilon^5$. So the probability that $\mathrm{WMajority}(\{w_{C_i}, \tilde{x}(C_i)\}_{x \in C_i, i \in [m]}) = -x^*$ is at most $\varepsilon^5$.

Suppose that the final output of $\phi_i$ consists of non-trivial-1 clauses with total weight $\alpha_i$, non-trivial-2 clauses with total weight $\beta_i$, satisfied clauses with total weight $\gamma_i$ and non-trivial-3 clauses with total weight $\zeta$, where $i \in \{1, 2\}$. The inequalities between them and $m'$ continue. Thus, by the same reasons, the final output of $\phi_2$ has the expected optimal assignment value at least $m' - \gamma_1$.

By Corollary 2.5, the subroutine WMAX3SATB$(\phi_i)$ has the same guarantee of approximation factor as the subroutine MAX3SATB$(\phi_i)$ of Algorithm 2. Therefore, the relevant calculations continue, indicating that Algorithm 4 finds an assignment with approximation factor at least $7/8 + \Theta(\varepsilon^2/\log(1/\varepsilon))$ in expectation. $\qquad\square$

# E   HARDNESS WITH VARIABLE SUBSET ADVICE

In this section, we complement the proofs of Theorem 1.6 and Theorem 1.7. Our main goal is to demonstrate the compatibility between the classic reductions and prediction-augmented framework. For the sake of proofs, we formalize some definitions as follows.

**Definition E.1.** *An algorithm $\mathcal{A}$ is a $(c, s)$-approximation algorithm if given a $c$-satisfiable formula, it finds a solution that satisfies at least an $s$-fraction of the constraints.*

**Definition E.2.** *Given an formula $\phi$, we denote by $\mathrm{Val}(\phi)$ the maximum fraction of constraints that can be satisfied by an assignment.*

For the sake of completeness, we state the Exponential Time Hypothesis (ETH) and Linear Size PCP Conjecture here. Our hardness results are under these conjectures.

**Conjecture E.3** (Exponential Time Hypothesis (Impagliazzo et al., 2001))**.** *There exists a constant $c \in (0, 1)$ such that for all large enough integers $n$, the 3SAT problem on $n$ variables cannot be solved in time $2^{cn}\mathrm{poly}(n)$.*

**Conjecture E.4** (Linear Size PCP Conjecture (Dinur, 2016))**.** *For some $C_1, C_2 > 0$ and all sufficiently small $\varepsilon > 0$, there exists a polynomial-time reduction from 3SAT to Label Cover that satisfies the following properties. Assume that the reduction maps a 3SAT instance $\phi$ of size $m$ to a Label Cover instance $\psi = (U, V, E, \sum_U, \sum_V, \{\pi_e\}_{e \in E})$. Then,*

- *$|U|, |V| \le (1/\varepsilon)^{C_1} \cdot m$.*

- *$|\sum_U|, |\sum_V| \le (1/\varepsilon)^{C_2}$.*

- *If $\mathrm{Val}(\phi) = 1$, then $\mathrm{Val}(\psi) = 1$.*

- *If $\mathrm{Val}(\phi) < 1$, then $\mathrm{Val}(\psi) \le \varepsilon$.*

## E.1   HARDNESS OF MAX-E3SAT

We obtain here the proof of Theorem 1.6. We mainly leverage two lemmas from (Ghoshal et al., 2025) and the reduction from Max-3-Lin to Max-E3SAT.

The first lemma is as follows. We actually need the Max-E3SAT version, which is an immediate corollary of the lemma.

**Lemma E.5** (Lemma 5.6 in (Ghoshal et al., 2025))**.** *Suppose there exists a polynomial-time algorithm $\mathcal{A}$ for MAX $r$-Lin that given a $c$-satisfiable formula $\phi$ and advice with parameter $\varepsilon$ in the Variable Subset Advice model, outputs a solution satisfying an $s$-fraction of the constraints with probability at least $0.9$ over the choice of the advice string. Then there exists a deterministic $(c, s)$-approximation algorithm $\mathcal{A}'$ for MAX $r$-Lin that runs in time $2^{(\varepsilon \log(4/\varepsilon))n}\mathrm{poly}(n)$.*

**Corollary E.6.** *Suppose there exists a polynomial-time algorithm $\mathcal{A}$ for Max-E3SAT that given a $c$-satisfiable formula $\phi$ and advice with parameter $\varepsilon$ in the Variable Subset Advice model, outputs a solution satisfying an $s$-fraction of the constraints with probability at least $0.9$ over the choice of the advice string. Then there exists a deterministic $(c, s)$-approximation algorithm $\mathcal{A}'$ for Max-E3SAT that runs in time $2^{(\varepsilon \log(4/\varepsilon))n}\mathrm{poly}(n)$.*

*Proof.* Since the proof of Lemma E.5 does not rely on specific properties of Max $r$-Lin but rather on general constraints satisfaction properties, it naturally extends to the broader class of CSP problems in the Variable Subset Advice model, we therefore obtain the analogous result of Max-E3SAT. $\qquad\square$

The second lemma involves standard complexity assumptions. We clarify that Lemma E.7 in (Ghoshal et al., 2025) states $\mathcal{I}$ consists of $2^{O(1/\varepsilon)^{C_3}}n$ constraints, but its proof states $2^{2(1/\varepsilon)^{C_3}}n$ constraints. This exact parameter is needed in our subsequent analysis. The following lemma can be proved by the techniques from (Håstad, 2001) under the assumption of ETH and linear-size PCP conjecture. A complete proof is provided in (Ghoshal et al., 2025).

**Lemma E.7** (Lemma 5.1 in (Ghoshal et al., 2025)). *Assume that the ETH and Linear Size PCP Conjecture hold. For some absolute constants $C_1, C_2, C_3 > 0$ and $\varepsilon_0 \in (0, 1/2)$, the following holds. For every $\varepsilon \in (0, \varepsilon_0)$ and $\eta(\varepsilon) = C_1/\sqrt{\log(1/\varepsilon)}$, there is no algorithm that given a Max 3-Lin formula $\mathcal{I}$ on $n$ variables and $2^{2(1/\varepsilon)^{C_3}}n$ constraints, distinguishes between the following cases:*

$$\textbf{Yes Case} : \mathrm{Val}(\mathcal{I}) \geq 1 - \eta(\varepsilon) \quad \text{and} \quad \textbf{No Case} : \mathrm{Val}(\mathcal{I}) \leq 1/2 + \eta(\varepsilon). \tag{6}$$

*in time $2^{2^{-(1/\varepsilon)^{C_2}}n} \cdot \mathrm{poly}(n)$.*

Based on the reduction from Max-3-Lin to Max-E3SAT, we can get the Max-E3SAT version of Lemma E.7 as follows.

**Lemma E.8.** *Assume that the ETH and Linear Size PCP Conjecture hold. For some absolute constants $C_1', C_2, C_3 > 0$ and $\varepsilon_0 \in (0, 1/2)$, the following holds. For every $\varepsilon \in (0, \varepsilon_0)$ and $\eta'(\varepsilon) = C_1'/\sqrt{\log(1/\varepsilon)}$, there is no algorithm that given a Max-E3SAT formula $\mathcal{S}$ on $n$ variables and $2^{2(1/\varepsilon)^{C_3}+2}n$ clauses, distinguishes between the following cases:*

$$\textbf{Yes Case} : \mathrm{Val}(\mathcal{S}) \geq 1 - \eta'(\varepsilon) \quad \text{and} \quad \textbf{No Case} : \mathrm{Val}(\mathcal{S}) \leq 7/8 + \eta'(\varepsilon). \tag{7}$$

*in time $2^{2^{-(1/\varepsilon)^{C_2}}n} \cdot \mathrm{poly}(n)$.*

*Proof.* Given a Max 3-Lin formula $\mathcal{I}$ on $n$ variables and $2^{2(1/\varepsilon)^{C_3}}n$ clauses, where $C_3$ is the constant in Lemma E.7. Based on $\mathcal{I}$, we can construct a Max-E3SAT formula $\mathcal{S}$ as follows:

1. $\mathcal{S}$ consists of $n$ variables in $\mathcal{I}$.

2. For any constraint $xyz = 1$ in $\mathcal{I}$, there are four clauses $(x \vee \bar{y} \vee \bar{z})$, $(\bar{x} \vee y \vee \bar{z})$, $(\bar{x} \vee \bar{y} \vee z)$ and $(x \vee y \vee z)$ in $\mathcal{S}$.

Here $x = 1$ means that $x$ is true. Then for any assignment of $n$ variables in $\mathcal{I}$, if $xyz = 1$ is satisfied, the four clauses are satisfied; otherwise, exactly one of the four clauses is not satisfied. Thus, $\mathcal{I}$ is $c$-satisfiable if and only if $\mathcal{S}$ is $(3+c)/4$-satisfiable. Following Lemma E.7, we get that there is no algorithm that given a Max-E3SAT formula $\mathcal{S}$ on $n$ variables and $2^{2(1/\varepsilon)^{C_3}+2}n$ constraints, distinguishes between the following cases:

$$\textbf{Yes Case} : \mathrm{Val}(\mathcal{S}) \geq 1 - \eta(\varepsilon) \quad \text{and} \quad \textbf{No Case} : \mathrm{Val}(\mathcal{S}) \leq 7/8 + \eta(\varepsilon)/4. \tag{8}$$

where $\eta(\varepsilon) = C_1/\sqrt{\log(1/\varepsilon)}$ and $C_1, C_2$ and $C_3$ are the constants in Lemma E.7.

To get the similar form as Lemma E.7, we can let $C_1' = C_1/4$ and $\eta'(\varepsilon) = C_1'/\sqrt{\log(1/\varepsilon)}$. Then the proof is finished. □

*Proof of Theorem 1.6.* Let $C_1'$ and $C_2$ be the constants in Lemma E.8. For every $\delta > 0$, let $\varepsilon_1 = \varepsilon_1(\delta) = 2^{-(C_1'/\delta)^2}$. By Lemma E.8, for any $\varepsilon \in (0, \varepsilon_1)$, there is no algorithm that decides whether a Max-E3SAT formula is at most $(7/8 + \delta)$ or at least $(1 - \delta)$-satisfiable in time $2^{2^{-(1/\varepsilon)^{C_2}}n} \cdot \mathrm{poly}(n)$. Define $\varepsilon_0$ such that $\varepsilon_0 \log(4/\varepsilon_0) \leq 2^{-(1/\varepsilon_1)^{C_2}}$. Now the theorem statement follows from Corollary E.6. □

### E.2  HARDNESS OF MAX-E3SAT(B)

We obtain here the proof of Theorem 1.7. We follow the same framework of proof as in the last subsection. The reduction needed is from Max-E3SAT to Max-E3SAT(B). Since the correctness of this reduction is not obvious, we provide a rewritten proof from (Trevisan, 2001) as a reference. The readers mainly need to know what the construction of reduction looks like.

**Corollary E.9.** *Suppose there exists a polynomial-time algorithm $\mathcal{A}$ for Max-E3SAT(B) that given a c-satisfiable formula $\phi$ and advice with parameter $\varepsilon$ in the Variable Subset Advice model, outputs a solution satisfying an s-fraction of the constraints with probability at least $0.9$ over the choice of the advice string. Then there exists a deterministic $(c, s)$-approximation algorithm $\mathcal{A}'$ for Max-E3SAT(B) that runs in time $2^{(\varepsilon \log(4/\varepsilon))n}\mathrm{poly}(n)$.*

*Proof.* We use the same argument as Corollary E.6. $\qquad\square$

**Theorem E.10** ((Trevisan, 2001)). *Let $\phi$ be an formula of Max-E3SAT. Let $B$ be a fixed and sufficiently large parameter. We can construct an formula of Max-E3SAT(B) that is denoted by $\phi_B$ such that if $\phi$ is not c-satisfiable, with high probability, then $\phi_B$ is not $c + 4/\sqrt{B}$-satisfiable.*

*Proof.* Given a Max-E3SAT formula $\phi$ on $n$ variables and $m$ clauses. Based on $\phi$, we can construct a Max-E3SAT(B) formula $\phi_B$ as follows:

1. For any variable $x$ in $\phi$, create a potential set $S_x = \{x_1, x_2, \ldots, x_{occ(x)}\}$.

2. Uniformly sample a clause $(x \vee y \vee z)$ in $\phi$. Then uniformly sample $x_i \in S_x$, $y_j \in S_y$ and $z_k \in S_z$. Add the clause $(x_i \vee y_j \vee z_k)$ into $\phi_B$.

3. Independently repeat the second step for $Bm/6$ times.

4. If there exists a variable $x_i$ that appears in more than $B$ clauses in $\phi_B$, then delete some clauses that contain $x_i$ until no such variable exists.

Clearly, $\phi_B$ is an formula of Max-E3SAT(B). Consider any variable $x_i$ in $\phi_B$. In one sampling step (one time whole second step), the probability that $x_i$ is sampled is $(occ(x)/m) \cdot (1/occ(x)) = 1/m$. So the expected number of occurrences of $x_i$ in $\phi_B$ before the deletion is $B/6$. By Heterogeneous Coin Flips, the probability that $x_i$ appears in $k \geq B$ clauses is at most $2^{-k}$. The expected number of the deleted clauses that contains $x_i$ is at most

$$\sum_{k=B}^{\infty}(k - B) \cdot 2^{-k} = 2^{-B}\sum_{i=0}^{\infty} i \cdot 2^{-i} = 2^{-B-1}\sum_{i=0}^{\infty} i \cdot 2^{-(i-1)} = 2^{-B+1} \leq 1. \tag{9}$$

where we use $\sum_{i=0}^{\infty} ix^{i-1} = \frac{d}{dx}\sum_{i=0}^{\infty} x^i = \frac{d}{dx}\frac{1}{1-x} = \frac{1}{(1-x)^2}$ for $|x| < 1$.

Since $\sum_x occ(x) = 3m$, $\phi_B$ consists of at most $3m$ variables. Then, the expected number of the deleted clauses is at most $m$. By Markov's Inequality, with probability at least $1 - 6/\sqrt{B}$, the number of the deleted clauses is at most $\sqrt{B}m/6$. We can replaced the deleted clauses by the trivial satisfied clauses to make calculations easier. We note that $\phi_B$ consists of $Bm/6$ clauses.

To analyze the relationship of satisfiabilities between $\phi$ and $\phi_B$, we need the following auxiliary weighted formula $\phi'$.

1. For any clause $(x \vee y \vee z)$ in $\phi$, there are $occ(x) \cdot occ(y) \cdot occ(z)$ clauses $(x_i, y_j, z_k)$ in $\phi'$, where $x_i \in S_x, y \in S_y, z \in S_z$ and $S_x, S_y, S_z$ are defined in the construction of $\phi_B$.

2. Each clause $(x_i, y_j, z_k)$ in $\phi'$ has the weight $1/(occ(x) \cdot occ(y) \cdot occ(z))$.

If $\phi$ is $c$-satisfiable, then clearly $\phi'$ is also $c$-satisfiable. Suppose that $\phi'$ has an assignment $A'$ that satisfies the clauses of total weights $cm$. We can consider the random assignment of any variable $x$ in $\phi$ where $x$ is assigned to True with probability proportional to the number of variables $\{x_i\}$ that are assigned to True in $A'$. For the random assignment where:

- an $\alpha$ fraction of variables $\{x_i\}$ are assigned False,

- a $\beta$ fraction of variables $\{y_i\}$ are assigned False, and

- a $\gamma$ fraction of variables $\{z_i\}$ are assigned False,

the fraction of unsatisfied clauses $\{(x_i \vee y_j \vee z_k)\}$ is $\alpha\beta\gamma$. Meanwhile, the probability that this random assignment makes $(x \vee y \vee z)$ unsatisfied is exactly $\alpha\beta\gamma$. Since the expected number of satisfied clauses in $\phi$ is $cm$, $\phi$ must have an assignment that satisfies the clauses of total weights at least $cm$. Therefore, $\phi$ is $c$-satisfiable if and only if $\phi'$ is $c$-satisfiable.

Suppose that $\phi'$ has an assignment $A'$ of the approximation factor $c$. So the probability that the clause sampled in one sampling step is satisfied by $A'$ is $c$. Let $M = Bm/6$ and $\varepsilon = 3/\sqrt{B}$. By Hoeffding's Inequality, the probability that more than $(c + \varepsilon)M$ of the sampled clauses are satisfied by $A'$ is at most $e^{-2\varepsilon^2 M} = e^{-m}$. Thus, if initial $\phi_B$ (before the substitution of trivial clauses) is $(c + \varepsilon)$-satisfiable, with probability at least $1 - e^{-m}$, $\phi'$ is $c$-satisfiable and $\phi$ is $c$-satisfiable. Using the negation we can get that if $\phi$ is not $c$-satisfiable, with probability at least $1 - e^{-m}$, initial $\phi_B$ is not $(c + 3/\sqrt{B})$-satisfiable, equivalently, there is no assignment for initial $\phi_B$ that satisfies more than $(c + 3/\sqrt{B})M$ clauses.

By the union bound, if $\phi$ is not $c$-satisfiable, with probability at least $1 - 6/\sqrt{B} - e^{-m}$, there is no assignment for $\phi_B$ that satisfies more than $(c + 3/\sqrt{B})M + \sqrt{B}m/6 = (c + 4/\sqrt{B})M$ clauses, equivalently, $\phi_B$ is not $(c + 4/\sqrt{B})$-satisfiable. □

Based on the reduction from Max-E3SAT to Max-E3SAT(B), we can get the Max-E3SAT(B) version of Lemma E.7 as follows.

**Lemma E.11.** *Assume that the ETH and Linear Size PCP Conjecture hold. Let $B$ be a fixed and sufficiently large parameter. For some absolute constants $C'_1, C_3, C_4 > 0$ and $\varepsilon_0 \in (0, 1/2)$, the following hold. For every $\varepsilon \in (0, \varepsilon_0)$ and $\eta(\varepsilon) = C'_1/\sqrt{\log(1/\varepsilon)}$, there is no algorithm that given a Max-E3SAT(B) formula $\mathcal{S}_B$ on $n$ variables and $Bn/18$ clauses, distinguishes between the following cases:*

$$\textbf{Yes Case}: \mathrm{Val}(\mathcal{S}_B) \geq 1 - \eta(\varepsilon) \quad \text{and} \quad \textbf{No Case}: \mathrm{Val}(\mathcal{S}_B) \leq 7/8 + 4/\sqrt{B} + \eta(\varepsilon). \tag{10}$$

*in time $2^{2^{-3(1/\varepsilon)^{C_4}} n} \cdot \mathrm{poly}(n)$.*

*Proof.* Given a Max-E3SAT formula $\mathcal{S}$ on $n'$ variables and $m = 2^{2(1/\varepsilon)^{C_3}+2} n'$ clauses, where $C_3$ is the constant in Lemma E.8. By Theorem E.10, we can construct a Max-E3SAT(B) formula $\mathcal{S}_B$ such that if $\mathcal{S}$ is not $c$-satisfiable, with high probability, then $\mathcal{S}_B$ is not $c + 4/\sqrt{B}$-satisfiable. To analyze the new parameters and new running time, we restate the construction in Theorem E.10 as follows:

1. For any variable $x$ in $\mathcal{S}$, create a potential set $S_x = \{x_1, x_2, \ldots, x_{occ(x)}\}$.

2. Uniformly sample a clause $(x \vee y \vee z)$ in $\mathcal{S}$. Then uniformly sample $x_i \in S_x$, $y_j \in S_y$ and $z_k \in S_z$. Add the clause $(x_i \vee y_j \vee z_k)$ into $\mathcal{S}_B$.

3. Independently repeat the second step for $Bm/6$ times.

4. If there exists a variable $x$ that appears in more than $B$ clauses, then delete some clauses that contain $x$ until no such variable exists.

We can see that $\mathcal{S}_B$ contains at most $\sum_x occ(x) = 3m$ variables and at most $Bm/6$ clauses. Let $n = 3m = 3 \cdot 2^{2(1/\varepsilon)^{C_3}+2} n'$. Then the number of clauses in $\mathcal{S}_B$ is at most $Bm/6 = Bn/18$, and the running time of the construction of $\mathcal{S}_B$ is $Bm/6 = Bn/18 = O(N)$. By Lemma E.8, the total running time is

$$2^{2^{-(1/\varepsilon)^{C_2}} n'} \cdot \text{poly}(n') + O(n) \leq 2^{2^{-(1/\varepsilon)^{C_2} - 2(1/\varepsilon)^{C_3} - 2} n - O(1/\varepsilon)^{C_3}} \cdot \text{poly}(n) \tag{11}$$

$$\leq 2^{2^{-3(1/\varepsilon)^{C_4}} n} \cdot \text{poly}(n). \tag{12}$$

where $C_2$ is the constant in Lemma E.8 and $C_4 = \min\{C_2, C_3\}$. $\qquad\square$

*Proof of Theorem 1.7.* Let $C_1'$ and $C_4$ be the constants in Lemma E.11. For every $\delta > 0$, let $\varepsilon_1 = \varepsilon_1(\delta) = 2^{-(C_1'/\delta)^2}$. By Lemma E.11, for any $\varepsilon \in (0, \varepsilon_1)$, there is no algorithm that decides whether a Max-E3SAT(B) formula is at most $(7/8 + 4/\sqrt{B} + \delta)$ or at least $(1 - \delta)$-satisfiable in time $2^{2^{-3(1/\varepsilon)^{C_4}} n} \cdot \text{poly}(n)$. Define $\varepsilon_0$ such that $\varepsilon_0 \log(4/\varepsilon_0) \leq 2^{-3(1/\varepsilon_1)^{C_4}}$. Now the theorem statement follows from Corollary E.9. $\qquad\square$

## F    HARDNESS OF MAX-E3SAT WITH CLAUSE ADVICE

In this section, to prove Theorem 1.5, we demonstrate the construction of that special Max-E3SAT instance and use the normal distribution to approximate the binomial distribution. Before the proof, we note that the same hardness applies to any algorithm which starts off similarly to ours, by plugging in values for variables that are very frequent (formalized in the proof of Theorem 1.5).

First we need a simple statement regarding the normal distribution.

**Lemma F.1.** *Suppose that $Z \sim \mathcal{N}(0, 1)$. For small $k > 0$, we have*

$$\Pr[Z \leq -k] \geq \frac{1}{2} - \frac{k}{\sqrt{2\pi}}.$$

*Proof.*

$$\Pr[Z \leq -k] = \Pr[Z \geq k] = \frac{1}{2} - \Pr[0 \leq Z \leq k] \geq \frac{1}{2} - k \cdot f(0) = \frac{1}{2} - \frac{k}{\sqrt{2\pi}}, \tag{13}$$

where $f(z) = \frac{1}{\sqrt{2\pi}} e^{-\frac{z^2}{2}}$ is the probability density function. $\qquad\square$

*Proof of Theorem 1.5.* We construct an unweighted formula of Max-E3SAT $\phi$ as follows:

1. $\phi$ only consists of variables that appear in the following clauses.

2. For $1 \leq i \leq m, 1 \leq j \leq n$, there are four clauses $(x_{ij} \vee y_{ij} \vee z_j)$, $(\overline{x_{ij}} \vee y_{ij} \vee z_j)$, $(x_{ij} \vee \overline{y_{ij}} \vee z_j)$ and $(\overline{x_{ij}} \vee \overline{y_{ij}} \vee z_j)$ in $\phi$.

We can see that any $x_{ij}$ or $y_{ij}$ appears in the 4 clauses, and any $z_j$ appears in the $4m$ clauses. Here, $m$ is used to bound the number of occurrences of each variable and $n$ is used to set the scale of $\phi$. Here we can see that all $z_j$ have the identical status (they are interchangeable in $\phi$). For simplicity of analysis, we can assume $n = 1$. Note that this means $z_1$ appears in all of the clauses.

If $z_1 = 0$, regardless of the value of $x_{i1}$ or $y_{i1}$ is, only three of the above four clauses are satisfied. If $z_1 = 1$, regardless of the value of $x_{i1}$ or $y_{i1}$, all the above four clauses are satisfied. When we randomly assign the value to $z_1$ without predictions, we cannot do better than the best classic $7/8$-approximation algorithm. Next we consider make use of the clause advice for only this very frequent variables $z_1$.

Let $Y$ be the random variable that represents the event where the majority prediction of $z_1$ is equal to True. The number of clauses that we satisfy if we set $z_1$ equal to its majority prediction is upper bounded by $4m \cdot \Pr[Y = 1] + 3m \cdot \Pr[Y = 0]$ and thus our expected approximation factor is given by

$$\frac{3 \cdot \Pr[Y = 1] + 4 \cdot \Pr[Y = 0]}{4}.$$

Set $m = 1/\varepsilon$. Let $p = (1 + \varepsilon)/2$ and define $X = \sum_{i=1}^{4m} X_i$ be the sum of the random variables $X_i$ such that $\Pr[X_i = 1] = p$ and $\Pr[X_i = 0] = 1 - p$ for any $1 \leq i \leq 4m$. Then $X \sim B(4m, p)$, where $B(4m, p)$ is the binomial distribution. Furthermore, $\Pr[Y = 1]$ is precisely $\Pr[X > 2m]$ and similarly, $\Pr[Y = 0]$ is $\Pr[X \leq 2m]$ (say in the event of a tie we vote for False. We can also randomize here and our conclusion will be quantitatively the same). Since $\varepsilon$ is sufficiently small, $m = 1/\varepsilon$ is sufficiently large. We can use the normal distribution $\mathcal{N}\big(4mp, 4mp(1 - p)\big)$ to approximate $B(4m, p)$ (up to an additive error going to 0 which we hide for simplicity). Let $Z = (X - 4mp)/\sqrt{4mp(1 - p)}$, then $Z \sim \mathcal{N}(0, 1)$.

By Lemma F.1, we have

$$\Pr[X \leq 2m] \approx \Pr[Z \leq -\frac{m(2p - 1)}{\sqrt{mp(1 - p)}}] = \Pr[Z \leq -4\sqrt{\frac{\varepsilon}{1 - 4\varepsilon^2}}] \geq \frac{1}{2} - \frac{4}{\sqrt{2\pi}}\sqrt{\frac{\varepsilon}{1 - 4\varepsilon^2}}.$$

Thus, we can find a class of algorithms in the Clause Advice model that only naturally leverage predictions, such that each of them has the expected approximation factor

$$\leq \frac{3 \cdot \Pr[X \leq 2m] + 4 \cdot \Pr[X \geq 2m]}{4} = \frac{4 - \Pr[X \leq 2m]}{4} \leq \frac{7}{8} + \frac{1}{\sqrt{2\pi}}\sqrt{\frac{\varepsilon}{1 - 4\varepsilon^2}} = 7/8 + O(\sqrt{\varepsilon}).$$

$\square$

# G EXPERIMENTS

We complement our theoretical results with an empirical evaluation for the Max-E3SAT problem. All of our experiments are conducted using Python on a M1 MacbookPro with 32GB of RAM.

**Experimental Setting** For our experiments, we also implement Hastad's algorithm given in Corollary 2.5 for input CSPs where every variable appears in a bounded number of instances. We believe this to be the first implementation of this algorithm, which could be of independent interest. We also implement a simplification of our main augmented algorithm, Algorithm 2. The simplification is that we don't implement line 4 in Algorithm 2 which calls the $7/8$ approximation algorithm of Karloff & Zwick (1997); Zwick (2002) for the case where clauses are not of equal size (but have at most 3 variables). Rather, we only take the best of $A_1$ and $A_2$ in line 3 of Algorithm 2. This is because the algorithms of Karloff & Zwick (1997); Zwick (2002) require running a complicated SDP which turned out to be infeasible in practice (even though they are

| CSP | # of Variables | # of Clauses |
|:---:|:---:|:---:|
| 1 | 50 | 80 |
| 2 | 50 | 100 |
| 3 | 50 | 100 |
| 4 | 50 | 170 |
| 5 | 100 | 160 |
| 6 | 100 | 200 |
| 7 | 100 | 340 |
| 8 | 200 | 680 |
| 9 | 200 | 1200 |

Table 1: Properties of CSPs used in our experiments.

theoretically polynomial time). For our algorithm, we report the average of 20 trials and shade one standard deviation.

All the CSP instances we use are satisfiable and all have only one optimal assignment. These instances are all generated with a particular Random-3-SAT instance generator Asahiro et al. (1996) and are downloaded from DIMACS Benchmark Instances - AIM. We use 9 CSPs and their properties are given in Table 1. For all CSPs, every variable appears in $\leq 18$ clauses.

We also consider two natural baselines. The first baseline is the main one, representing the classic $7/8$ approximation algorithm which is optimal assuming $P \neq NP$. Note that it's expected approximation ratio is exactly $7/8$ for any input and it does not look at the structure of the input CSP at all. The second baseline is Hastad's algorithm from Corollary 2.5. Note that it is not a general algorithm since its guarantees require the input to have a reasonable 'bounded' occurrence for every variable (which the inputs do not meaningfully satisfy). Nevertheless, we found it has strong performance in practice as discussed below.

Our algorithm requires a setting of the high-degree threshold (see line 1 in Algorithm 2). Theoretically, this value should decreases as $\epsilon$ increases. In practice, we pick a simple scaling by initially picking $B = 10$ in all cases and decreasing it by 1 for every $0.1$ increase in $\epsilon$.

**Results** We again note that the main baseline we are comparing to is the classic $7/8$ approximation factor shown in red. Our results are shown in Figure 1. As we vary $\epsilon$ (note $\epsilon$ ranges from 0 to 1; see Definition 1.3), our algorithm consistently outperforms the classic $7/8$ approximation baseline across all ranges of $\epsilon$. This validates our theoretical guarantees of Theorem 1.4 experimentally. Surprisingly, Hastad's algorithm also performs very well (even though theoretically it only has strong guarantees under the case that $B$ is bounded in Corollary 2.5 ). Nevertheless, as $\epsilon$ (which corresponds to the quality of the prediction) increases, the learning-based algorithms eventually outperforms both baselines and its approximation factor approaches 1. This holds true across all of the 9 CSPs. Thus, our algorithm displays robustness (the performance is always as good as the classic $7/8$ approximation for all ranges of $\epsilon$) and consistency (the performance improves as $\epsilon \to 1$).

Note that some of the curves display a 'step function' behavior (e.g. the top row of Figure 1). This is because we discretely decrease the value of the 'high degree' threshold $B$ in our algorithm (which is an integer parameter). For these inputs, the setting of $B$ has a high impact in the performance of our algorithm. Thus, we can view these curves as regimes where Hastad's algorithm dominates and where predictions dominate, although the relationship is not so straightforward to analyze since we only apply Hastad's algorithm after simplifying the CSP using predictions (e.g. on the output of Algorithm 1). For these plots, our algorithm is also able to follow the strong performance of Hastad even for very low values of $\epsilon$, but as $\epsilon$ increases, the algorithm is more reliant on the predictions (this has to be the case since $\epsilon \to 1$ should result in perfect

approximation), leading to a 'phase transition' in some of the approximation curves in Figure 1. This can be intuitively thought as the point where the predictions start 'dominating' the algorithm.

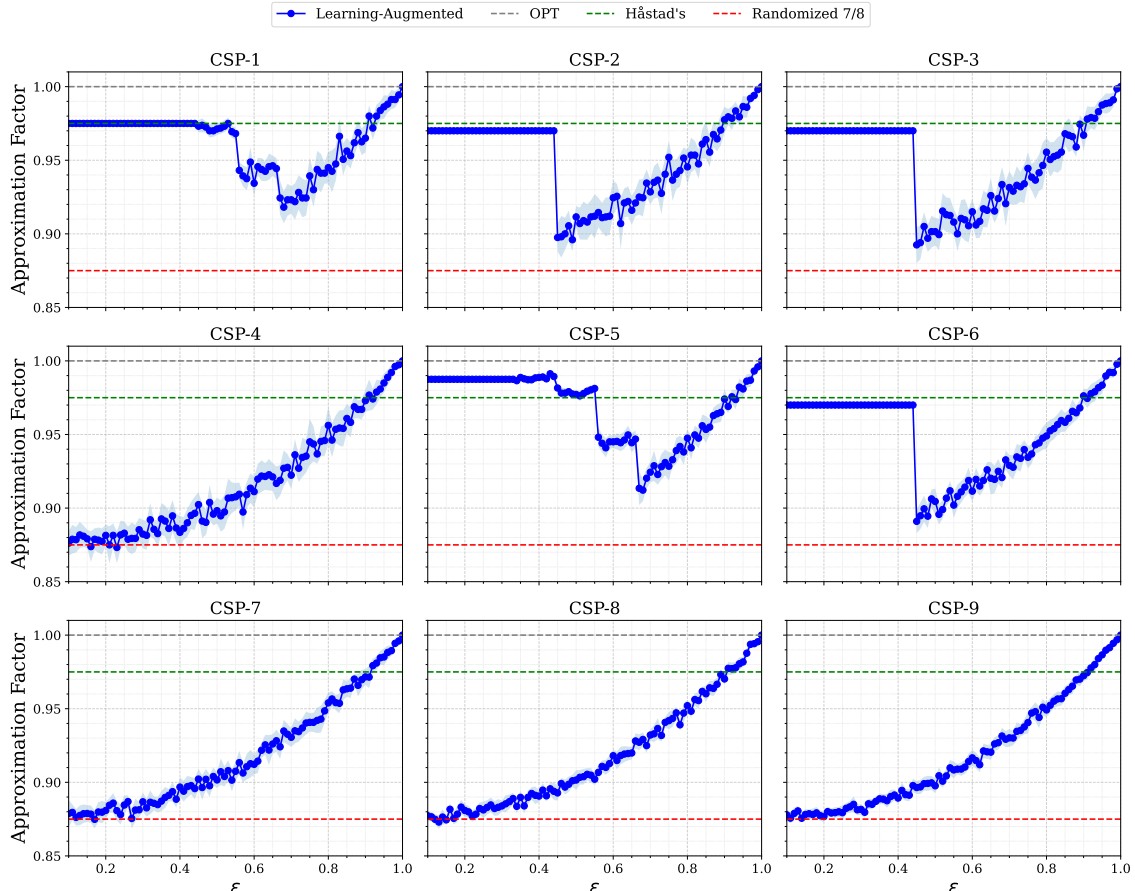

Figure 1: Approximation Factor vs $\epsilon$ for nine different CSP instances. Each subplot corresponds to a different CSP-$i$. Horizontal lines represent: OPT (gray), Håstad's algorithm of Corollary 2.5 (green), and the classic 7/8 randomized baseline (red), which is our main baseline.

