# OpenReview forum: "Using Clause Predictions for Learning-Augmented Constraint Satisfaction"
_ICLR.cc/2026/Conference — Submitted to ICLR 2026_

### Official Review · Reviewer_s6SC · 2025-10-30

**Soundness:** 4
**Presentation:** 4
**Contribution:** 4
**Rating:** 8
**Confidence:** 4

**Summary:**

This paper studies Max-E3SAT with learning-augmented predictions. It introduces the Clause Advice model where each clause provides noisy sign predictions for its variables. The main result is a polynomial-time algorithm achieving an approximation ratio that surpasses the classic 7/8 barrier. The algorithm combines majority decoding for high-occurrence variables with bounded-occurrence techniques. Hardness results show ohter prediction models cannot beat 7/8, and an upper bound establishes limitations of majority-based strategies.

**Strengths:**

Clause Advice model ties predictive power to variable importance. Inverted-predictions construction is a clever hedging strategy.

Complete proofs including detailed re-derivation of bounded-occurrence techniques.

Clear writing.

 Breaks known complexity barrier

**Weaknesses:**

Independence claim for certain events relies on disjointness of constructed sets. This could be stated more explicitly.

**Questions:**

Could alternative decoding strategies potentially beat the upper bound limitation?

---

> ### Author Response · Authors · 2025-11-21
> **Reply to Reviewer s6SC**
>
> We thank the reviewer for their careful reading and comments. We address their questions below:
>
> > Independence claim for certain events relies on disjointness of constructed sets. This could be stated more explicitly.
>
> Thank you for the detailed comment regarding the proof of Theorem 1.4 (if we understand exactly where you are pointing out). We note that the independence of the events $E(x_j)$ is inherited from the independence of $\tilde{C_i}$ (as defined by the clause advice). Furthermore, we use the disjointness of constructed sets $S_{x_j}$ in the subsequent counting step.
>
> > Could alternative decoding strategies potentially beat the upper bound limitation?
>
> We expect that if clause advice model follows a distribution consistent with the intuition that variables appearing more frequently are easier to predict accurately, even without full independence, our algorithm should continue to operate effectively, because this assumption affects only a small aspect of our algorithm (Line 3 in Algorithm 1), while the remaining components are agnostic to it.

---

### Official Review · Reviewer_gHxa · 2025-10-30

**Soundness:** 3
**Presentation:** 3
**Contribution:** 2
**Rating:** 6
**Confidence:** 3

**Summary:**

This study explores the problem of MAX-E3SAT within the context of learning-augmented algorithms, utilizing oracle advice. The MAX-E3SAT problem involves a conjunctive normal form (CNF) formula where each clause consists of exactly three literals. The objective is to find a Boolean assignment that maximizes the number of satisfied clauses. After examining traditional prediction models—label advice and subset advice—the authors introduce a novel prediction model called "clause advice." In this model, each clause can be represented by a vector of bits corresponding to the values of the literals in a given assignment. Essentially, each clause provides a noisy variant of the vector associated with the optimal assignment. Within this framework, each bit in the clause is modeled as a Bernoulli random variable that offers a slightly more accurate prediction than a random uniform choice.

Using $\epsilon$ to represent this information, the authors develop a $ \frac{7}{8} + \Theta(\epsilon^2 / \log(1/\epsilon)) $-approximation algorithm, that recovers the classical random choice algorithm when $\epsilon$ tends to zero. They also establish a hardness result for this scenario. Additional approximation and hardness results are provided for the problem MAX-E3SAT[$B$], where each variable occurs in at most $B$ clauses.

**Strengths:**

**S1.** The paper is well-written, and the notation used is clear. The technical aspects are easy to follow, even for readers who are not experts in learning-augmented algorithms. The results are presented in a pedagogical manner, effectively explaining their significance and the main ideas behind the proofs.

**S2.** Although the study specifically addresses the MAX-E3SAT problem, the approximation and hardness results extend beyond mere extensions of existing findings. Therefore, the “clause advice” model could also be utilized to address other MAX-CSP problems.

**Weaknesses:**

**W1.** While the approximation and hardness results are not straightforward extensions of existing findings, they are limited to the specific MAX-E3SAT problem. Their practical implication is therefore questionable.

**W2.** Although the "clause advice" model is conceptually interesting, it relies on the assumption that each "literal occurrence" in a clause behaves like an independent Bernoulli random variable. This assumption feels somewhat counterintuitive, as the assignment of any literal in a given constraint naturally affects its value in other constraints. Therefore, I am not entirely convinced that this predictive model is more appropriate than the classic "label advice" model.

**W3.** While the authors present approximation and hardness results for both the "clause advice" model and the "subset advice" model, they do not provide similar results for the natural "label advice" model. The authors conjecture that the $\frac{7}{8}$ bound for this oracle cannot be improved, but it would be helpful to include some informative comments regarding this conjecture.

**Questions:**

This study focuses on the MAX-E3SAT problem. The authors provide a brief discussion of extensions in Section 3. However, I am particularly interested in whether the results can be applied to variants of MAX-SAT. Here are my questions:

**Q1.** Can we extend the approximation and hardness results to the standard MAX-3SAT problem, where each clause contains at most three literals?

**Q2.** Is it possible to improve the approximation bound in the clause advice model when the instances of MAX-E3SAT are randomly generated?

**Q3.** Although the MAX-2SAT problem has already been studied within the context of the "label advice" model, can we achieve similar approximation and hardness results for the "clause advice" model?

---

> ### Author Response · Authors · 2025-11-21
> **Reply to Reviewer gHxa**
>
> We thank the reviewer for their careful reading and comments. We address their questions below:
>
> > W1: " ... they are limited to the specific Max-E3SAT problem. Their practical implication is therefore questionable."
>
> We study Max-3SAT as an example that we can use more powerful clause advice to achieve some improvement when the normal label or subset advice cannot work. It is reasonable to infer that some more difficult CSP problems are more likely to be solved by clause models than by other two models. Furthermore, we have **added a new section in the appendix** (Appendix G) presenting a proof of concept experimental results comparing our augmented algorithm against baselines for Max-E3SAT. In the experiments, our augmented algorithm displays robustness (the performance is always as good as the classic 7/8 approximation for all ranges of $\epsilon$) and consistency (the performance improves as $\epsilon \to 1$). This demonstrates the practicality of our algorithm.
>
> > the assumption that each "literal occurrence" in a clause behaves like an independent Bernoulli random variable. This assumption feels somewhat counterintuitive
>
> The independence assumption reflects the intuition that variables appearing more frequently are easier to predict accurately. Our clause-advice model formalizes this intuition. Importantly, this assumption affects only a small aspect of the algorithm (Line 3 in Algorithm 1), while the remaining components are agnostic to it. We expect that if clause advice model follows a distribution consistent with this intuition, even without full independence, the algorithm should continue to operate effectively.
>
> > they do not provide similar results for the natural "label advice" model.
>
> We would like to clarify this point with the reviewer. As detailed in the paragraph preceding Theorem $1.6$, our statement is precisely that "any hardness result for the Variable Subset Advice automatically applies to the Label Advice". Specifically, given variable subset advice, it is easy to simulate label advice: simply keep the predictions for the variables given by the variable subset advice and randomly choose an assignment for the rest of the variables.
>
> > The authors conjecture that the bound for this oracle cannot be improved
>
> Our Theorem $1.6$ shows that for any fixed approximation factor $\delta > 0$, there exists some sufficiently small accuracy factors $\varepsilon > 0$ for which the problem remains hard (NP-hard). It is close to the desired hardness result, but not tight. So we expect a more rigorous result to confirm our conjecture.
>
> > Can we extend the approximation and hardness results to the standard Max-3SAT problem
>
> We confirm that the algorithms and techniques developed in our study are readily applicable to the standard Max-3SAT problem. However, the reason we emphasized Max-E3SAT is because our worst-case performance guaranty relies on a parameter $c=7/8$ (as shown in Corollary $2.7$). However, for the standard Max-3SAT problem, we only have $c\geq 3/4$, which is insufficient to guarantee breaking the classical barrier.
>
> > Is it possible to improve the approximation bound in the clause advice model when the instances of Max-E3SAT are randomly generated?
>
> We strongly agree with the reviewer's intuition. It is very possible to improve the approximation bound for randomly generated Max-E3SAT instances. Our current analysis focuses on the worst-case scenario, which, by design, has a relatively low probability of occurring in a standard randomly generated instance.
>
> > Although the Max-2SAT problem has already been studied within the context of the "label advice" model, can we achieve similar approximation and hardness results for the "clause advice" model?
>
> Thank you for the question. Given the significant differences between the classical research approaches for Max-2SAT and Max-3SAT, we cannot directly transfer the insights developed for Max-3SAT in the Clause Advice model to Max-2SAT. However, Investigating Max-2SAT within the Clause Advice Model is indeed is an interesting future research direction to identify other constraint satisfaction problems that can benefit from the clause prediction model that we introduce.

---

> > ### Comment · Reviewer_gHxa · 2025-11-26
> >
> > I thank the authors for their detailed response. I particularly appreciate the clarification regarding MAX-3SAT. I also recognize that MAX-2SAT is quite different and warrants a separate paper. Finally, I believe that the proof of concept presented in Appendix G is a valuable addition.

---

### Official Review · Reviewer_aXid · 2025-11-01

**Soundness:** 3
**Presentation:** 1
**Contribution:** 2
**Rating:** 4
**Confidence:** 2

**Summary:**

An approximation factor of 7/8 is provably the optimal for polynomial algorithms for MAX-3SAT. To improve on this, learning-augmented algorithms utilizes additional information (advice). Existing approaches include Label Advice and Variable Subset Advice models. However, in MAX-3SAT, Label Advice is believed to give no improvement, while Variable Subset Advice makes the problem too easy to improve by assuming knowledge of exact assignments without uncertainty. This paper introduces Clause Advice models where a clause level predictor that makes noisy prediction where its correct with probability $(1+\epsilon)/2$ is available. The paper then proves that there exist a polynomial-time algorithm in Clause Advice model that can find an assignment with approximation factor lower bounded by $7/8 + \theta(\epsilon^2 / log(1/\epsilon))$ in expectation.

**Strengths:**

The proposed algorithm builds on prior work by incorporating the concept of degree for the variables.

The clause advice model is a simple yet intuitive extension to the label advice model for 3SAT.

Extensive proofs are provided for the theorems.

**Weaknesses:**

The paper is difficult to follow and can use some reorganization. The theorems are presented early on without context, and related contents are scattered throughout the paper. The presentation can also be improved, for example, the main algorithm is described in one paragraph on the top of page 4. A simple example can clarify the algorithm much better.

The clause advice model’s applicability beyond Max-3SAT remains to be demonstrated and may be less general across CSPs than label or subset advice.

It is difficult to gauge the significance of the improved bounds of the proposed model.

The method assumes the variable predictions are independent across clauses, which is intuitively less likely in practice.

**Questions:**

Line 88 seems to suggest that Label, Variable Subset and Clause Advice are the only three natural prediction models. Is that intended? What about Clause Advice models that use predictions on the clause as a whole instead of the variables in the clauses?

---

> ### Author Response · Authors · 2025-11-21
> **Reply to Reviewer aXid**
>
> We thank the reviewer for their careful reading and comments. We address their questions below:
>
> > A simple example can clarify the algorithm much better.
>
> We have added concrete examples in Section 2.1 illustrating how predictions guide our algorithm and why they improve performance. The change is highlighted in blue to make it easy to see, which will be removed in the final version.
>
> > The clause advice model’s applicability beyond Max-3SAT remains to be demonstrated and may be less general across CSPs than label or subset advice.
>
> While clause advice is indeed more specialized than label or subset advice, it enables qualitatively stronger results in settings where it applies. In particular, for Max-3SAT it allows us to surpass the classical 7/8 barrier, something not achievable with the existing advice models.
>
> > It is difficult to gauge the significance of the improved bounds of the proposed model.
>
> Our work pinpoints specific barriers faced by existing prediction models for Max-3SAT. By introducing clause advice and surpassing the classical $7/8$ barrier (which holds if P is not equal to NP), we provide the evidence that more expressive prediction models can yield strictly stronger guarantees. This highlights the potential of prediction-augmented algorithms for Max-CSPs.
>
> > The method assumes the variable predictions are independent across clauses, which is intuitively less likely in practice.
>
> The independence assumption reflects the intuition that variables appearing more frequently are easier to predict accurately. Our clause-advice model formalizes this intuition. Importantly, this assumption affects only a small aspect of the algorithm (Line 3 in Algorithm 1), while the remaining components are agnostic to it. We expect that if clause advice model follows a distribution consistent with this intuition, even without full independence, the algorithm should continue to operate effectively. Furthermore, we rely on concentration bounds for proving concentration of certain error terms. These should be easily replaceable with O(1)-wise independence, but we believe this is not technically interesting.
>
> > "Line 88 .... Is that intended?"
>
> This is not intended. While label advice and variable-subset advice provide meaningful improvements in many learning-augmented settings, they are fundamentally limited for Max-3SAT: some hardness results exist. This limitation motivates our introduction of clause advice.
>
> > What about Clause Advice models that use predictions on the clause as a whole instead of the variables in the clauses?
>
> If clause advice were limited to predicting only the value of the entire clause, it would provide essentially no useful information for Max-3SAT. For example, consider a dominant variable that appears in every clause. After removing this variable, the remaining variables across clauses may be mutually inconsistent (e.g., $x\vee y$, $\bar{x}\vee y$, $x\vee \bar{y}$ and $\bar{x}\vee \bar{y}$). If we assign the wrong value to the dominant variable, we may only be able to satisfy $3/4$ of the clauses. The current clause advice model can effectively handle such a situation by capturing the dominant variable.
>
> Please let us know if we have satisfactorily addressed your concerns and we are happy to continue the discussion further!

---

### Official Review · Reviewer_w6JR · 2025-11-04

**Soundness:** 3
**Presentation:** 1
**Contribution:** 3
**Rating:** 4
**Confidence:** 2

**Summary:**

This paper studies the Max-E3SAT problem—an NP-hard constraint satisfaction problem—and introduces a new Clause Prediction Model within the learning-augmented algorithms framework. In this model, each clause provides a noisy bit of information (accurate with probability 1/2 + \epsilon) about the optimal assignment of its variables. Using this model, the authors design a polynomial-time algorithm that achieves an approximation ratio of 7/8 + \Theta(\epsilon^2 / \log(1/\epsilon)) surpassing the classical 7/8 barrier known to be optimal in the worst case (under P ≠ NP).
Overall, the paper advances the theoretical understanding of how noisy predictions can help solve NP-hard problems beyond worst-case limits. The paper is technically solid but difficult to follow for non-specialists, and no experimental validation or discussion of real-world applicability is included, which may reduce the perceived relevance for ICLR.

**Strengths:**

1.	Introduces a new, natural, and noise-tolerant advice model (“clause predictions”) that is well-motivated and distinct from existing label or subset advice frameworks.
2.	Solid theoretical analysis with rigorous proofs and clear complexity-theoretic hardness arguments.
3.	Highlights that “per-constraint” (clause-level) predictions can be more informative than “per-variable” predictions, offering a new research direction for learning-augmented combinatorial optimization.

**Weaknesses:**

1.	The paper is very difficult to read for researchers not already familiar with approximation algorithms or CSP theory. The exposition jumps quickly into formal definitions with minimal intuition or examples.
2.	The core idea—why clause predictions help, and how they allow going beyond 7/8—is mathematically clear but conceptually opaque. The link between prediction accuracy \epsilonε, variable degree, and approximation improvement is not well visualized or intuitively explained.
3.	The writing style mimics theoretical computer science papers (tight, notation-heavy, and formal), not the more expository tone expected at ICLR.
4.	No experimental validation or discussion of real-world applicability is included, which may reduce the perceived relevance for ICLR.

**Questions:**

1.	Can you provide more intuition or a simple example illustrating how clause predictions improve approximation beyond 7/8?
2.	How sensitive is the algorithm’s performance to adversarial noise beyond the assumed independence in the clause predictions?

---

> ### Author Response · Authors · 2025-11-21
> **Reply to Reviewer w6JR**
>
> > The exposition jumps quickly into formal definitions with minimal intuition or examples.
>
> We have added additional intuition in the introduction to better motivate the setting before presenting formal definitions and Question 1. The PDF has been updated with the changes highlighted in blue (which will be removed in the final version).
>
> > why clause predictions help, and how they allow going beyond 7/8.
>
> We have added concrete examples in Section 2.1 illustrating how predictions guide our algorithm and why they improve performance.
>
> > The writing style mimics theoretical computer science papers (tight, notation-heavy, and formal), not the more expository tone expected at ICLR.
>
> Ultimately our paper is about a theoretical contribution to constraint satisfaction problem in the learning-augmented model. Our intention was to follow the expository style common to theoretically focused works, with intuition and high-level explanations placed in the main text and technical details deferred to the appendix. We believe the current structure maintains accessibility while keeping the theoretical arguments rigorous.
>
> > No experimental validation or discussion of real-world applicability is included
>
> We have **added a new section in the appendix** (Appendix G) presenting a proof of concept experimental results comparing our augmented algorithm against baselines for Max-E3SAT. In the experiments, our augmented algorithm displays robustness (the performance is always as good as the classic 7/8 approximation for all ranges of $\epsilon$) and consistency (the performance improves as
> $\epsilon \to 1$). Note that this is to be expected since we mathematically proved that this behavior holds.
>
> > How sensitive is the algorithm’s performance to adversarial noise beyond the assumed independence in the clause predictions?
>
> Thank you for the question. Intuitively, adversarial noise may cause clause predictions to misidentify the optimal assignment for "high degree" variables. However, our *inverting* steps ($\phi_2$, Algorithm $1$) can naturally correct the effect of a \textit{single} adversarial noise, especially when that noise influences multiple clauses in the final assignment.  A quantitative analysis of more complex or multiple noise patterns is not yet understood, and investigating such richer noise models would be a very interesting direction for future work. However, note that we can never do worse than the classical $7/8$ approximation since we can always run the classical algorithm in parallel and take the best answer (which can be easily checked).
>
> We believe we have addressed your main concerns about adding more intuition/examples as well as a simple empirical evaluation. Please let us know if we have satisfactorily addressed your concerns and we are happy to continue the discussion further!

---

### Author Response · Authors · 2025-11-25
**Response to reviewers**

Dear reviewers,

We have updated the document to include more intuitive description of our method (see the text in blue), as well as proof of concept experiments, showing that our theoretical algorithm displays both consistency and robustness. We are happy to engage further with the reviewers as well.

Many thanks,
The authors

---

### Author Response · Authors · 2025-11-30
**Summary of changes to AC**

Dear AC,

Thank you for your hard work. We wanted to provide a summary about the updates we have made during the rebuttal. The reviewers had concerns about the theoretical natural of the paper. Indeed, it is true that our main focus is on a fundamental constraint satisfaction problem and how predictions can help us achieve a better theoretical guarantee, beyond worst-case performance. This is aligned with the field of learning-augmented algorithms. Furthermore, we agree that the main contribution of the paper is algorithmic and theoretical in nature.

In the rebuttal phase, we have added more intuitive, high level exposition throughout the paper (in blue), as well as a proof of concept experimental section, showing that our algorithm displays better performance when the predictions are accurate (consistency), while retaining the classic approximation guarantee when predictions are noisy (robustness).

Many thanks,
The authors

---

### Meta-Review · Area_Chair_tsF1 · 2026-01-05

**Summary:**

The paper presents an improved approximation algorithm for MAX-E3SAT under a clause advice model (Definition 1.3). They note that unlike previous advice models (Definitions 1.1, 1.2), quantitative improvements appear possible while technically non-trivial. The main result is Theorem 1.4, which gives the improved approximation, complemented with some qualitatively-similar conditional lower bounds.

The reviewers appreciated the technical contribution, but also raised concerns about how realistic the setting is, and the relevance to an ICLR audience. The most positive review was also the most lacking in substance. The primary concerns raised were re: independent advice per clause, specificity to the MAX-E3SAT problem, and relatively incremental improvement (along with practical relevance).

I think the main conceptual idea is reasonably innovative (get advantage on common variables + use restricted solver on the rest), but I agree with the reviewers that the contribution may not be very aligned with ICLR. Definition 1.2 led to a rather trivial prior algorithm, but "it's more of a technical challenge" is not compelling enough justification for a new model. I suggest the authors either focus on motivating a realistic model (ideally, with experiments), or emphasize the theoretical novelty. Re: the latter, the theoretical contributions do not seem actionable enough for ICLR (e.g., they are at best a small constant factor improvement over the baseline), and may be better-appreciated at a venue with more focus on constraint satisfaction algorithms.

**Reviewer Concerns:**

I think the exposition did improve after the reviewer comments.

I think the motivation of the "advice" setting, even as a toy theoretical model, is still relatively weak. Perhaps the authors could carry out their conjectured generalization under a more robust condition, e.g., some sort of weaker advice that provides nontrivial advantage on "heavy" variables? For an ICLR audience, I think one compelling end-to-end application where the advice model is reasonable, and the approximation improvement is significant (even if the ultimate CSP instance evaluated is a toy model) would be quite helpful. For a theory conference, highlighting the technical novelty and potential consequences for related problems might be the best next step.

**Reviewer Scores:**

I think that some of the reviewers' writing concerns may be alleviated, but the concerns re: how relevant or realistic the setting is, and how appropriate the paper's fit is for ICLR, would remain.

---

### Decision · Program_Chairs · 2026-01-26

Reject